# UV Sensitivities of Two Littoral and Two Deep-Freshwater Amphipods (Amphipoda, Crustacea) Reflect Their Preferred Depths in the Ancient Lake Baikal

**DOI:** 10.3390/biology13121004

**Published:** 2024-12-02

**Authors:** Elizaveta Kondrateva, Anton Gurkov, Yaroslav Rzhechitskiy, Alexandra Saranchina, Anastasiia Diagileva, Polina Drozdova, Kseniya Vereshchagina, Zhanna Shatilina, Inna Sokolova, Maxim Timofeyev

**Affiliations:** 1Institute of Biology, Irkutsk State University, Irkutsk 664025, Russia; lizzarium@gmail.com (E.K.); a.n.gurkov@gmail.com (A.G.); alexandra147801@gmail.com (A.S.); dyagilevanastasia@gmail.com (A.D.); drozdovapb@gmail.com (P.D.); k.p.vereshagina@gmail.com (K.V.); zhshatilina@gmail.com (Z.S.); 2Baikal Research Centre, Irkutsk 664003, Russia; 3Department of Marine Biology, Institute for Biological Sciences, University of Rostock, 18059 Rostock, Germany; 4Department of Maritime Systems, Interdisciplinary Faculty, University of Rostock, 18059 Rostock, Germany

**Keywords:** amphipods, arthropods, Baikal, crustaceans, crustacyanin-like proteins, deep sea, reductive evolution, UV-screening compounds

## Abstract

Any species that does not experience certain harmful conditions for an extended period can lose the corresponding protective mechanisms. The search for the organisms with such losses in unusual environments is important to decipher these mechanisms and better understand evolution. Global climate change influences aquatic ecosystems in multiple ways, including shifting water transparency. The latter determines how deep solar ultraviolet can penetrate waterbodies and which species are favored by the change since ultraviolet is lethal for many organisms. Thus, understanding the effects of ultraviolet is important for the management of aquatic ecosystems. Protection against ultraviolet is energetically costly and is known to be reduced in organisms dwelling in dark environments, but most studied species are marine. Lake Baikal is the only freshwater reservoir, where animals adapted to dark deep-water zones in parallel with oceanic counterparts, which makes them unique objects to study parallel evolution. Here, we show that scavenger amphipods (shrimp-like crustaceans) inhabiting different parts of the Baikal deep-water zone partially lost their ultraviolet protection proportionally to preferred depths. Their ultraviolet tolerance was found to be related to the concentration of ultraviolet-screening compounds such as carotenoids.

## 1. Introduction

Solar ultraviolet (UV) radiation is among the most significant ecological factors influencing life on Earth’s surface. It is classified into several types, with UV-A (400–315 nm) and UV-B (315–280 nm) being the most environmentally relevant [1,2]. Absorption of the more energetic UV-B, unlike UV-A, is highly dependent on the amount of atmospheric ozone [3,4], which is one of the motivations for studying the biological effects of UV-A and UV-B separately. The direct influence of UV on terrestrial ecosystems is obvious, but both UV-A and UV-B can also penetrate the upper layers of aquatic ecosystems. The water transparency for UV greatly varies, but, in the waterbodies with the lowest concentrations of dissolved organic matter, a 10% reduction of UV intensity is found down to depths of at least 30 m for UV-A and 15 m for UV-B [5,6,7].

The roles of UV radiation in aquatic ecosystems are tightly bound to a multitude of other factors in complex ways, and both higher or lower UV levels reaching certain depths can shift the animal communities by favoring different species [7,8]. This fact requires a thorough understanding of UV adaptation mechanisms and their past evolutionary changes in various taxonomic groups in order to improve predictions and management of changes in aquatic ecosystems. UV adaptation strategies generally include avoidance, protection with UV-screening compounds such as melanin or carotenoids, and coping with UV effects on the cellular level if the first two options are ineffective or impossible [9]. The outcomes of UV exposure for animal cells include direct DNA damage and enhanced production of reactive oxygen species (ROS) that may trigger a counteracting increase in the activities of enzymatic and non-enzymatic antioxidants [9,10].

The ancient Lake Baikal located deep in Eurasia provides unique opportunities for studying the evolution of UV adaptation mechanisms. Its water is one the most transparent of all lakes in the world [11]. Most importantly, Baikal is the birthplace of the only deep-water fauna arisen in freshwater, which inhabits the lake down to its maximal depths of over 1640 m [12,13]. In the case of highly diverse endemic amphipods (Amphipoda, Crustacea) descending from shallow-water *Gammarus*-like species [14], the species diversification involved adaptations to high UV levels in the littoral zone of the lake as well as gradual colonization and adaptation to the expanding deep-water zone with no UV at all [15]. Thus, we can hypothesize that modern Baikal littoral amphipods should be well adapted to UV, while deep-water species could have lost their systems for UV protection. Such loss of energetically expensive protection systems was found, for example, in the case of the response to rising temperature in Antarctic notothenioid fish [16]. Since deep-water amphipods of Lake Baikal emerged independently from their oceanic counterparts [14], the evolutionary changes in their adaptation mechanisms, particularly those related to UV protection, hold significant value for understanding both the evolution of UV defense and the processes that lead to the loss of adaptive traits when selective pressures diminish. Besides evolutionary significance, UV intolerance of deep-water animals might have ecological implications for predicting their migration patterns if the lake water becomes less transparent for solar UV.

The littoral zone of Lake Baikal is inhabited by a plethora of species, many of which belong to the genus *Eulimnogammarus* [17]. Among the most widespread species of the genus, *E. cyaneus* (Dybowsky, 1874) is one of the smallest and *E. verrucosus* (Gerstfeldt, 1858) is one of the largest (Figure 1) and, thus, may demonstrate different sensitivities to UV. The diversity of amphipods in the dark deep-water zone of Baikal is also high. The representatives of the deep-water genus *Ommatogammarus* are particularly useful as a model for comparative studies, as these nektobenthic scavengers [18] are easy to collect using traps with rotten fish bait and are similar in size to the littoral species (Figure 1). Recently, we showed that *O. flavus* (Dybowsky, 1874) and *O. albinus* (Dybowsky, 1874) inhabiting wide ranges of depths can both be used in experiments under atmospheric pressure [19], which is essential for comparisons with littoral species. *O. albinus* has greater preferred depths than *O. flavus* and paler coloration, which potentially indicates the higher sensitivity of the former to UV.

Here, we aimed to estimate the tolerance of the deep-water Baikal amphipods under environmentally relevant UV levels in comparison to the littoral species and explore the biochemical mechanisms underlying their differences. We observed a decrease in UV tolerance with increasing preferred depth in the two studied *Ommatogammarus* species and found a link to the differences in the levels of carotenoids and carotenoid-binding proteins in these amphipods. Additionally, we tested the ability of the deep-water species to prey on the littoral amphipods without the pressure of solar UV radiation to evaluate their potential to invade more shallow-water communities if the lake water becomes less transparent.

## 2. Materials and Methods

### 2.1. UV Measurements

The intensities of the UV radiation were mostly measured separately for UV-A (main probe sensitivity is in the range of 315–400 nm and residual sensitivity extends down to 300 nm and up to 420 nm) and UV-B (main probe sensitivity is in the range of 282–315 nm and residual sensitivity extends down to 270 nm and up to 325 nm) using TKA-PKM(13) UV radiometer (TKA, Saint Petersburg, Russia). Specifically for underwater UV measurements (combined UV-A and UV-B; the probe is sensitive in the range of 260–395 nm), we used a Center 532 UV radiometer (Center, Taipei, Taiwan) fixed inside a glass container (IKEA, Malmö, Sweden) mounted on a metal frame to keep the UV probe horizontally underwater. The intensity of the visible light was always measured with a Megeon-21170 luxmeter (Megeon, Moscow, Russia). In the case of solar visible light, the measured illuminance was converted to irradiance with the formula 1 lux = 0.0079 W/m^2^ [22].

We monitored the intensities of solar UV radiation and visible illumination near the shoreline of Lake Baikal from 4 to 10 July 2024 in the Bolshie Koty village (51.90325° N 105.06981° E) at 6 a.m., 12 p.m., 3 p.m., 6 p.m., and 9 p.m. The probes were always directed to the sun during the measurements. Underwater UV measurements at 1 m depth were performed at the same location on 9 July 2022 at about 2 p.m. when the sky was totally clear.

Visible and UV output of all light sources used for the experiments was estimated on the distance of 25 cm from the lamps, which is the distance between the lamps and the bottom of the aquaria in the used experimental setup.

### 2.2. Animal Sampling and Maintenance

All experimental procedures with amphipods were conducted in accordance with the EU Directive 2010/63/EU for animal experiments and were approved by the Animal Subjects Research Committee of the Institute of Biology at Irkutsk State University (protocol #009). All studied species are neither endangered nor protected. The species identification was conducted using the taxonomic keys of Bazikalova [20].

Adults of the littoral amphipods *E. cyaneus* and *E. verrucosus* were collected with a hand net near the shoreline of Lake Baikal at the Listvyanka village (51.87058° N 104.82827° E) from depths of 0–1 m during summer and early autumn (if not stated otherwise). The deep-water scavengers *O. flavus* and *O. albinus* were sampled using 5-l plastic traps with rotten fish in net bags as bait. The traps were installed from ice in March and placed at the bottom of the bay of Bolshie Koty village at depths of approximately 300, 600, and 700 m, as described previously [19]. The difference in sampling times for littoral and deep-water amphipods was due to seasonal conditions in Lake Baikal and forced us to perform the most of further analyses on two pairs of species separately.

The amphipods were sorted by species under temperature-controlled conditions, transported to the laboratory in insulated boxes, and kept separately by the species in 2 L glass tanks (bottom size 12 × 15 cm) filled with continuously aerated water from Lake Baikal. The littoral *Eulimnogammarus* species were kept at 6 °C (average temperature in the littoral zone of Lake Baikal [23]) and acclimated to laboratory conditions for 4–7 days, while deep-water *Ommatogammarus* species were kept at 4 °C (except for the comet assay, when the experiment temperature was also 6 °C due to technical restrictions) and acclimated for at least 1 but no more than 4 weeks. Water exchanges with the following feeding were performed every 3 days. For littoral species, the feed was the dried and ground mixture of algae and amphipods from the sampling location [24], while in the case of deep-water scavengers, we used pieces of white fish meat [19].

### 2.3. Experimental Design for UV Exposures

Amphipods were subjected separately to two types of exposures with the leading influence of either UV-A or UV-B. We used two lamps TL44D25/09N (Philips, Amsterdam, The Netherlands) per aquarium for UV-A treatments or two lamps TB218-12 combined with one lamp TB218-10 (Simple Zoo, Moscow, Russia) per aquarium for UV-B treatments. All used UV lamps also had a substantial visible output, and control groups of amphipods were subjected to visible illumination without UV using either one daylight luminescent lamp (Philips) for control to UV-A or two lamps TB218-02 (Simple Zoo) for control to UV-B. The control lamps were wrapped into dark plastic films of different transparency to roughly match the illuminance of the respective experimental lamps and to completely remove the remaining UV. During the experiments, all lamps gradually lost their intensity, which we noticed only at the end of the study. At the beginning of the study, the outputs of the lamps were the following: 5.2 W/m^2^ of UV-A and 0.16 W/m^2^ of UV-B for the UV-A treatment, and ~0.4 W/m^2^ of UV-B and ~1 W/m^2^ of UV-A for the UV-B treatment. After completion of the study, the outputs decreased to the following: 1.6 W/m^2^ of UV-A and 0.05 W/m^2^ of UV-B for the UV-A treatment, and 0.1 W/m^2^ of UV-B and 0.4 W/m^2^ of UV-A for the UV-B treatment. The total visible illuminance of experimental and control lamps, respectively, at the end of the study was ~300 lx and ~500 lx for the UV-A treatment, as well as ~1.2 klx and ~1.6 klx for the UV-B treatment. The distance from the lamps to the bottom of the tanks, where the animals prefer to stay, was 25 cm.

All UV exposures consisted of 24 h cycles including 12 h of illumination and 12 h of darkness. Most experiments (except for the comet assay) consisted of four aquaria (each is considered an independent replicate) per UV treatment per species, as well as four aquaria for the respective UV-A or UV-B control treatments with only visible illumination. The height of the water column was always 12 cm, and no stones were provided in the aquaria for the UV exposure to prevent the animals from hiding under them.

The list of analyzed parameters from each experiment was slightly different in each case. For littoral species, we performed two separate sets of 10-day-long experiments for estimation of their survival under UV after separate samplings (i.e., one set was performed sequentially on *E. cyaneus* and *E. verrucosus* and another set was carried out sequentially on these species at a different time). For deep-water amphipods, only one 10-day-long experimental set was performed to estimate their survival under UV (sequentially on *O. albinus* and then on *O. flavus*). UV-A and UV-B exposures were always installed in parallel. Thus, the total number of UV-A treatment replicates was 8 for each littoral species and 4 for each deep-water species, and the same for the control conditions to UV-A and 2 groups of the UV-B experiments. During these experiments, each aquarium contained 20 individuals of *E. cyaneus*, 10 (for UV-A exposures) or 15 (for UV-B exposures) individuals of *E. verrucosus*, 10 individuals of *O. flavus*, or 10 individuals of *O. albinus*. Mortality of amphipods was checked once a day. In the survival experiments on *Ommatogammarus* species, we also included an additional parallel control group that experienced darkness during the whole experiment (two aquaria per UV type per species; see Appendix A). Since there were no statistically significant differences in survival (see Section 2.9 for the procedure description) between the main control groups with visible illumination and the additional control groups with no illumination for both species and both types of control lamps (all four *p* > 0.2; χ^2^ ≤ 1.6), the latter groups are not depicted.

The first experimental set for survival estimation on littoral species was also used for the analysis of amphipod locomotor activity, and the surviving individuals were shock-frozen in liquid nitrogen for measuring the antioxidant enzyme activities (the individuals from the second set were not used further). During the survival experiments on deep-water species, we also analyzed their locomotor activity but further biochemical measurements were impossible due to high mortality. Thus, specifically for the enzymatic assays we performed another set of 3-day-long experiments on *Ommatogammarus* scavengers with the same design as described above.

The comet assay was applied to the hemocytes of amphipods from additional sets of experiments for all species. In this case, 2–3 animals of either species per aquarium (one aquarium per species at a time) were subjected to UV-A or UV-B for 3 days and used for the comet assay immediately after the exposures. No parallel control group was installed but the initial control group was used instead. The experiments were repeated 4–6 times for each species.

### 2.4. Estimation of Locomotor Activity

The horizontal locomotor activity of amphipods was recorded using a Tough TG-5 camera (Olympus, Beijing, China) on days 1, 2, 3, 4, 6, 8, and 10 of the UV exposures. The video recording was 5 min long and was performed 5–6 h after the start of the illumination on the specific day. The activity was estimated as the distance covered by the individual in the horizontal plane over time. In the ideal case, we chose at least 5 random individuals per aquarium and tracked their movements for 1 min (also chosen randomly during the recording) in ImageJ software v1.53q [25] using the MTrackJ plugin. However, in specific cases, the number of analyzed individuals could be reduced from at least 20 in total due to poor visibility of individual animals (amphipods sometimes aggregate in one corner of the aquarium) or high mortality, especially for *Ommatogammarus*.

### 2.5. Enzymatic Assays

After exposure to UV radiation, the frozen amphipods were used for measuring the activities of antioxidant enzymes peroxidase (POD), catalase (CAT), and glutathione-S-transferase (GST) using standard spectrophotometric methods [26]. In total, 3 individuals of *E. cyaneus*, 1 individual of *E. verrucosus*, and 1–2 individuals of *Ommatogammarus* amphipods were used per sample for measurement of all enzymes from each sample. The extraction of enzymes was carried out in 0.1 M sodium phosphate (NaP) buffer with pH 6.5 using a 1:4 w:v sample-to-buffer ratio. Enzyme activities were measured at 25 °C using a Carry 50 UV/visible spectrophotometer (Varian, Palo Alto, CA, USA). The total peroxidase activity in the soluble fraction was measured using 4.4 mM guaiacol as the substrate in NaP buffer with pH 5 at 436 nm [27]. Catalase activity was measured with 2.25 mM hydrogen peroxide as the substrate in NaP buffer with pH 7 at 240 nm [28]. The glutathione-S-transferase activity was measured with 0.97 mM 1-chloro-2,4-dinitrobenzene as the substrate in NaP buffer with pH 6.5 at 340 nm [29]. Enzyme activities were normalized per concentration of total protein measured using the Bradford method [30].

### 2.6. Comet Assay

We also estimated the amount of double- and single-stranded breaks in hemocyte DNA with comet assay. These cells were chosen due to their easy extraction procedure. Immediately after UV exposures, hemolymph was sampled by the puncture of the amphipod exoskeleton between the sixth and seventh segments, as described previously [31], without mixing the extracted hemolymph with an anticoagulant buffer. One individual of either species was used per sample. Comet assay was performed according to the protocol adapted for *Drosophila* with electrophoresis in an alkaline buffer cooled to 4 °C [32]. DNA was visualized after staining with ethidium bromide in the RFP channel of an inverted fluorescent microscope Celena S (Logos Biosystems, Anyang-si, Republic of Korea) with a 10× objective, and the obtained images were further analyzed using specialized software CaspLab v1.2.3b2 comparing the ratio between damaged and undamaged DNA in each cell. Each biological replicate included at least 140 cells. The results are presented as tail moment (TM), i.e., the product of the tail length and percentage of DNA in the tail among all nuclear DNA [33,34].

### 2.7. Estimation of Carotenoids and Carotenoid-Binding Protein

Adult individuals of *E. verrucosus*, *O. flavus*, and *O. albinus* were sampled in the early spring of the same year to estimate the concentrations of a carotenoid-binding protein in hemolymph and carotenoids in the rest of the body. *E. cyaneus* was omitted from this analysis due to the small size of this species leading to difficulties in sampling enough amounts of hemolymph. Amphipods were acclimated in the laboratory, their hemolymph was extracted as described above (Section 2.6) and they were immediately frozen in liquid nitrogen. Total carotenoid content in individual amphipods was assessed with a spectrophotometry-based method as described previously [35] using a Carry 50 UV/visible spectrophotometer (Varian, USA).

The extracted hemolymph was immediately mixed 1:1 v:v with the anticoagulant buffer [31], followed by 1:1 v:v dilution with the 2× Laemmli application buffer [36] and incubation at 95 °C for 5 min before further storage at −20 °C. An amount of 10 μL of diluted hemolymph samples (i.e., containing 2.5 μL of native hemolymph) from all three species were subjected to one-dimensional polyacrylamide gel (10%) electrophoresis and stained with Coomassie Brilliant Blue. The procedure was performed as described previously [31] except for the number of animals used per hemolymph sample: 1 individual of *E. verrucosus*, 1 individual of *O. albinus*, and 2–3 animals for *O. flavus*. Stained gels were scanned jointly, and the bands corresponding to the 25 kDa carotenoid-binding protein [35] were quantified in ImageJ [25] after converting the image to gray values and calibrating the pixel values to optical density using a built-in function (option “Uncalibrated OD”). Optical density was calculated both for original images and the “background” images obtained using the function “Subtract background” (with the option to “Create background”), and the latter values were subtracted from the former in order to obtain the final concentrations of the carotenoid-binding protein in arbitrary units.

### 2.8. Cohabitation Experiments with Littoral and Deep-Water Amphipods

To estimate the predatory or competitive activity between the littoral and deep-water amphipods, we kept 30 individuals of *O. flavus* with either 60 individuals of *E. cyaneus* or 30 individuals of *E. verrucosus* for 3 weeks at 6 °C after samplings in early spring. Only *O. flavus* was used in these experiments, as it is better adapted to atmospheric pressure than *O. albinus*. The 2:1 density ratio of *E. cyaneus* to *O. flavus* was used due to the smaller size of the former species (Figure 1). We also used *E. verrucosus* with a length of 1–1.5 cm in order to roughly match the size of *O. flavus*. For this experiment, we used two large aquaria with a bottom size of 45 × 36 cm for each mixed group and covered the bottom with stones of various sizes to let the animals hide. The amphipods experienced a dim light regime without UV, and their mortality was checked every five days during the whole experiment.

### 2.9. Data Analysis and Statistics

All data were analyzed in R v4.3.1 [37] mostly with built-in functions and visualized using the ggplot2 v3.4.4 [38] and ggbeeswarm v0.7.2 packages. Coefficients of determination of the linear regression models were estimated using the lm and summary functions. Visualization of survival curves, their statistical comparisons, and estimation of median lethal times (LT_50_) was performed with the packages survival v3.5-7 [39] and survminer v0.4.9 [40]. The survdiff function was used for the pair-wise comparisons between survival curves, and the resulting *p*-values are always reported throughout the study without correction for multiple comparisons since the figure presents one comparison per axis. Additionally, correcting for all seven comparisons per UV treatment (i.e., four “experiment vs. control” comparisons presented in the figure, as well as two “main control vs. additional control” comparisons for deep-water species and one “experiment vs. experiment” comparison between two deep-water species mentioned only in the text) would gain the same conclusions but produce overcorrected *p*-values for the littoral species.

Statistical significance of differences in measured parameters under UV exposures was estimated in comparison to respective parallel or initial (only in the case of comet assay) control groups using the Mann–Whitney test with Holm’s correction for multiple comparisons separately within each facet of the figure panel. All possible comparisons within facet were performed with the same procedure only in the case of contents of carotenoids and carotenoid-binding protein in three species. The non-parametric test was chosen for all parameters due to relatively small sample sizes, and the procedure was implemented with the functions wilcox.test and p.adjust. The differences were considered statistically significant with corrected *p* < 0.05.

## 3. Results

### 3.1. Natural UV Illumination over Shoreline and in the Littoral Zone of Lake Baikal

We monitored UV intensity over the Lake Baikal shoreline at the beginning of July in order to place our experiments in an ecologically relevant context (Figure 2a,b). The highest measured irradiances were ~40 W/m^2^ and ~2.5 W/m^2^ for UV-A and UV-B respectively during this week-long monitoring. Taking into account all the data obtained at the sunniest part of the day (here measured at 12 p.m. and 3 p.m.) under both clear and cloudy skies, the median irradiance was ~18 W/m^2^ and ~1 W/m^2^ for UV-A and UV-B, respectively. UV-A showed a non-linear dependence on visible sunlight irradiance (Figure 2c; adjusted *R*^2^ = 0.79) with UV-A intensities increasing disproportionally at high visible illumination. On the contrary, UV-A and UV-B relations were more linear (Figure 2d; adjusted *R*^2^ = 0.97).

We also managed to roughly estimate the reduction in the combined UV-A and UV-B irradiance at 1 m depth at the same location where the monitoring was performed. The estimate was only relative since the probe was always located horizontally and not directed to the sun and covered with thick glass reducing UV-A and UV-B by 11% and 48%, respectively. On the day of measurements, the UV reduction at 1 m depth at the littoral zone was 36.5 ± 2.3% in comparison to the shoreline level (Appendix A).

The maximal UV outputs of the lamps chosen for the laboratory experiments were the following: 5.2 W/m^2^ of UV-A for the UV-A treatment (also gave 0.16 W/m^2^ of UV-B) and ~0.4 W/m^2^ of UV-B for the UV-B treatment (also gave ~1 W/m^2^ of UV-A). These values are 3.5 and 2.5 times lower than the median UV-A and UV-B irradiances measured during the sunny part of the day (Figure 2a,b).

### 3.2. Survival and Locomotor Activity of Baikal Amphipods Under UV Radiation

Since both the UV-A and UV-B lamps have substantial illumination in the visible part of the spectrum, we used daylight lamps to expose the control groups of animals. Survival of littoral Baikal amphipods *E. cyaneus* and *E. verucosus* (Figure 3) was identical in experimental and control conditions both under UV-A and UV-B treatments (all four comparisons: *p* ≥ 0.06; χ^2^ ≤ 3.7), and their median mortality never reached 20% during the 10-day-long experiments. On the contrary, deep-water species *O. flavus* and *O. albinus* (Figure 3) demonstrated statistically significant differences between the experimental and control groups in both types of UV exposure (all four comparisons: *p* << 0.001; χ^2^ ≥ 20.5). Moreover, *O. flavus* showed longer survival than *O. albinus* in both experiments with an estimated LT_50_ of 8.5 versus 5 days under the UV-A treatment and 8 versus 4 days under the UV-B treatment. The greater mortality of *O. albinus* under both UV types is also supported by the statistically significant differences in direct comparisons between the survival curves of two deep-water species (both comparisons: *p* < 0.001; χ^2^ ≥ 12.9). Median mortality in the control groups of two deep-water species reached 20% only for *O. albinus* on day 7 under UV-A (the experiment was terminated for the species on this day due to high mortality in the experimental group) and was lower in all other cases during 10 days of exposures.

Since stones were not provided as a potential cover for the animals during the exposures, we were able to monitor their horizontal locomotor activity in these experiments (Figure 4). The only species affected by UV-A in comparison to control conditions was littoral *E. cyaneus*, which statistically significantly decreased locomotor activity under UV (>5-fold by median compared to the control group) from the 6th day (*p* < 0.001; *U* = 31.5) until the end of the 10-day-long experiment (*p* = 0.012 and 0.012 for days 8 and 10; *U* = 86.5 and 12). *E. cyaneus* also showed statistically significant changes in locomotion under the UV-B treatment, but these were weaker and more variable: a 1.6–1.7-fold decrease on the 3rd (*p* < 0.001; *U* = 126) and 4th (*p* = 0.013; *U* = 239) days and a 2.3-fold increase on 6th day (*p* = 0.002; *U* = 588) compared with the control group.

On the contrary, exposure to the UV-B treatment caused a pronounced response for littoral *E. verrucosus* and deep-water *O. flavus* (Figure 4b). Statistically significant decreases in locomotor activity were observed for *E. verucosus* on the 4th (*p* = 0.005; *U* = 116) and 8th (*p* = 0.043; *U* = 108) days resulting in 13- and 3-fold decreases in the median speed of locomotion relative to the control. For *O. flavus*, the median speed of locomotion decreased by > 20-fold on the 6th (*p* < 0.001; *U* = 70) and 8th (*p* = 0.004; *U* = 78) days of exposure. Deep-water *O. albinus* showed no statistically significant response in locomotor activity to UV-A or UV-B radiation compared with the parallel control group but its activity was generally low.

### 3.3. Oxidative Status and DNA Damage in Baikal Amphipods Under UV Radiation

To assess the oxidative status of the animal tissues, we measured the activities of antioxidant enzymes (Figure 5). The littoral amphipods were exposed to UV for 10 days before the measurements, while the exposure time for deep-water species was shortened to 3 days due to the high mortality (Figure 3). Statistically significant changes were found only for GST activity of deep-water species (Figure 5c). In particular, UV-A caused a reduction of GST activity (a 2.4-fold decrease in median) in *O. flavus* (*p* < 0.001; *U* = 63), while UV-B led to a 2.1- and 1.6-fold increase in the median GST activity both for *O. flavus* and *O. albinus* compared with the control groups (*p* < 0.001 and *p* = 0.016; *U* = 0 and 18).

To assess the effects of UV radiation on the DNA damage in circulating hemocytes, we applied the comet assay after three days of UV exposure to all species (Figure 6). Statistically significant differences from the initial control group were found only for deep-water *O. flavus* with increases in the median amount of DNA breaks in hemocytes by 2.7 times under UV-A and by 5.6 times under UV-B (both *p* < 0.001; *U* = 30 and 22).

### 3.4. Carotenoids and Carotenoid-Binding Protein in Baikal Amphipods

The concentrations of UV-protective carotenoids and 25 kDa carotenoid-binding protein (CBP) [35] were measured, respectively, in the body and the hemolymph of three species—*E. verrucosus*, *O. flavus* and *O. albinus*. The littoral species *E. verrucosus* contained 3.2 times more carotenoids than deep-water *O. flavus* (*p* = 0.002; *U* = 48) and 8.2 times more carotenoids than deep-water *O. albinus* (*p* = 0.002; *U* = 36) by median (Figure 7a). The difference between two deep-water amphipods also was statistically significant with 2.6 times higher median carotenoid content in *O. flavus* than in *O. albinus* (*p* = 0.002; *U* = 48). The levels of CBP in hemolymph of these species followed a similar pattern (Figure 7b), with median CBP concentration in littoral *E. verrucosus* being 2.5 and 4.1 times higher than in *O. flavus* and *O. albinus*, respectively (both *p* = 0.007; both *U* = 36). The difference in CBP content between the two deep-water species was not statistically significant (*p* = 0.31; *U* = 25).

### 3.5. Interaction of Deep-Water O. flavus with Littoral Species

Since deep-water amphipods showed high susceptibility to natural UV levels, we were interested in how they could compete with the littoral species in shallow water if this factor was excluded. Thus, we tried to model the long-term interaction of *O. flavus* being housed either at 1:2 density with smaller *E. cyaneus* or at 1:1 density with *E. verrucosus* of comparable size under no UV radiation (Figure 8).

Cohabitation of *O. flavus* and *E. cyaneus* resulted in moderate mortality of both species reaching 20% and 8%, respectively, after three weeks of the experiment (Figure 8). Oppositely, mixing *O. flavus* and *E. verrucosus* in the same large aquarium led to a high mortality of *O. flavus* over 50% with just 6% mortality for the littoral species.

## 4. Discussion

In this study, we aimed to assess the UV sensitivity of adult endemic amphipods from littoral and deep-water habitats of Lake Baikal. Our research showed species- and habitat-specific differences in the survival, locomotion, and DNA damage under UV irradiation, as well as in the levels of the potential UV-protective components like carotenoids and an antioxidant enzyme. One limitation of this study was the gradual decrease in UV intensities of the used lamps, which was not immediately detected and led to declining levels of UV exposure throughout the study. While this limitation complicates the exact determination of the UV dose received by experimental animals, the applied ranges of UV intensities are well below the median intensities of solar ultraviolet on the water surface during the day and, thus, are environmentally relevant for the Baikal littoral zone.

Our estimates of the UV intensity at 1 m depth of Baikal water showed that approximately 64% of incoming UV radiation reached this depth. Assuming that most of the UV radiation measured at 1 m depth was UV-A and considering the median UV-A intensity measured at the surface during the sunny part of the day, the applied UV-A levels at the beginning and the end of our study corresponded to ~3 m and ~5 m water depth, respectively. In the case of applied UV-B intensities, we could not estimate the corresponding depths.

We first conducted the exposures to estimate survival, locomotor, and enzymatic activities on *E. cyaneus* and *E. verrucosus* and only after that start the same experiments for *O. flavus* and *O. albinus*. Importantly, both deep-water species were always processed for each parameter at the same time, which allows for direct comparison of their mortality. The experiments for the comet assay were the latest and performed for all four species in parallel. Since two littoral species experienced higher UV output than the two deep-water species, the observed differences in mortality between them emphasize the considerably higher sensitivity of the deep-water species to UV radiation.

Overall, we found no UV intolerance in *E. cyaneus* and *E. verrucosus* that dwell from the shoreline and down to over 10 m depth. In contrast, *O. flavus* and *O. albinus* were more vulnerable to UV radiation, with their sensitivity increasing in correlation with the greater preferred depths (100–200 m and 300–500 m for *O. flavus* and *O. albinus*, respectively [19]). Notably, *O. flavus* was occasionally reported from 2.5 m depth, whereas the pale *O. albinus* was not found above 47 m depth [20]. These findings indicate that UV radiation might be a contributing factor in limiting the vertical distribution of the deep-water Baikal amphipods, particularly of the more UV-sensitive *O. albinus*. Comparative studies on aquatic animals showed that species-specific or individual-specific UV sensitivities can be found in microscopic to large organisms such as fish and even whales [8,41], but the majority of research on UV effects has been performed on relatively small objects up to several millimeters in size such as zooplankton and early developmental stages of larger species [10,42,43,44]. Juveniles of sizes similar to the here studied *Ommatogammarus* adults were analyzed for two marine fish species with different preferred depths and showed substantially different survival under UV [45], similar to our results. Relatively large invertebrates are studied less in this context, but some attention is drawn to them in polar regions due to low solar UV, especially under ice [42], which resembles deep-water zones by this parameter. In particular, some studied polar amphipods were found to differently tolerate UV, again, in agreement with their spatial distribution [46,47,48]. Solar UV might be limiting the distribution of cave amphipods to open waterbodies, but the reported results on this topic are very preliminary [49]. Deep-sea amphipods are very diverse with many of them having pale coloration [50] and, therefore, are expected to also be UV intolerant. However, we were unable to find any studies checking this theory in practice, probably, due to more difficult procedures of sampling and handling such animals than in the case of Lake Baikal.

The species-specific differences in amounts of carotenoids and the carotenoid-binding protein (CBP) correlated with the observed differences in survival of *E. verrucosus*, *O. flavus*, and *O. albinus* under UV radiation. Although here we could not estimate other important UV-screening compounds such as mycosporines and mycosporine-like amino acids, these findings are consistent with the well-established role of carotenoids as protective pigments against UV radiation [9]. However, the two estimated parameters correlated with each other and, here, we could not distinguish the effects of free and protein-bound carotenoids for UV protection. The species-specific differences in the carotenoid system may suggest an evolutionary loss of UV protectors in response to the long-term absence of solar UV radiation, particularly in *O. albinus*. However, the exact biochemical and genetic mechanisms driving this loss, along with the potential for ontogenetic plasticity, remain to be explored.

Our motivation for monitoring amphipod locomotor activity under UV was the search for avoidance reactions and elevated activity. On the contrary, we mostly observed the negative effects of UV irradiation on the locomotor activity of Baikal amphipods, suggesting UV-induced behavioral stress. Interestingly, *E. cyaneus* exhibited a decrease in activity under UV-A irradiation, whereas *E. verrucosus* did not, which may indicate higher UV sensitivity in *E. cyaneus*, potentially due to its smaller size. While *O. flavus* and *O. albinus* were generally less active than littoral species, *O. flavus*, along with *E. cyaneus* and *E. verrucosus*, showed a reduction in locomotor activity under UV-B irradiation. The ecological significance of these locomotor disturbances remains unclear, but they could potentially impair the ability of littoral amphipods to distribute for long distances without a shelter protecting them from solar UV. This may contribute to geographic isolation of genetic lines within littoral species by sandy parts of the Baikal shoreline [51].

Measurements of the antioxidant enzymes peroxidase and catalase did not indicate oxidative stress under UV irradiation in any of the four amphipod species studied. Importantly, more direct markers of elevated ROS production such as lipid peroxidation products or protein carbonylation were not estimated in this study and are necessary to draw the final conclusions. Glutathione-S-transferase (GST) activity similarly showed no statistically significant increase in the UV-exposed littoral species. However, interestingly, in both deep-water species exposed to UV-B, we observed a notable increase in GST activity. GST is a multifunctional enzyme that plays an important role in the detoxification of exogenous and endogenous toxic compounds [52,53]. Thus, here, we did not find signs of significant ROS production in the studied amphipod species under the selected UV levels but UV-B specifically may cause accumulation of some toxic compounds in the less UV-tolerant deep-water species. However, considering that UV-A penetrates to greater depths than UV-B [5], exposure to UV-B radiation is likely ecologically irrelevant for *O. albinus* and may be only partially relevant for *O. flavus*.

Interestingly, we observed hemocyte DNA damage only in *O. flavus*, while no such damage was detected in *O. albinus* or the littoral species. In the comet assay experiments, all species were exposed to the same dose of UV radiation but the used procedure did not include DNA pre-treatment with any endonucleases before the assay. Thus, we could not detect cyclobutane pyrimidine dimers that mostly appear under UV and could only find double- and single-stranded breaks [54]. The absence of a UV response in *O. albinus* is challenging to explain, but we propose two possible hypotheses. First, hemocytes are primarily located deep within the organism, where they are shielded by outer tissues. In *O. albinus*, this protection could be further enhanced by a higher lipid content [55]. Additionally, since *O. albinus* is less adapted to atmospheric pressure, it is plausible that immune functions were generally suppressed, leading to lower gene expression in hemocytes and more condensed chromatin, which might be less vulnerable to UV-induced DNA breaks. Lastly, we cannot rule out survivor bias, where individuals of *O. albinus* with better DNA protection or repair mechanisms may have preferentially survived, reducing the observed DNA damage response. However, the enzyme-modified comet assay should shed some more light on this topic.

Among the factors analyzed, UV-B appeared to have the most pronounced effects on the four amphipod species we studied. Nevertheless, there are a couple of caveats: (i) we could not fully exclude UV-A from the UV-B lamps and UV-B from the UV-A lamps and (ii) the applied UV-B was slightly more intense than UV-A compared to their natural median levels, which were 2.5 and 3.5 times lower than their observed medians during monitoring. Additionally, the cumulative UV dose plays a critical role in UV effects [9], and, in our study, the 12 h exposures without hiding spots likely subjected the amphipods to a higher UV dose than they would naturally encounter in the littoral zone. Thus, we cannot rule out the possibility that UV exposures more closely mimicking the natural daily light cycle might result in slower mortality rates for deep-water amphipods. A notable drawback of this study was the decline in the lamp intensities, which was due to extensive use of the setup without regular checks of its parameters. As so, the use of such lamps would usually require exposing groups of different species strictly in parallel to allow direct comparison, while for the species-oriented experimental design as presented here, more stable LED sources seem to be more appropriate.

Overall, our findings suggest that solar UV radiation is one of the ecological factors limiting the distribution of deep-water amphipods to the littoral and possibly sublittoral zones of Lake Baikal. Observations from deep-water traps, where *Ommatogammarus* scavengers were found feeding on other species (as indicated by their remains), prompted us to investigate their potential to prey on littoral species under atmospheric pressure and without UV exposure. This experiment effectively modeled a scenario of reduced water transparency in the littoral zone, which could occur, for example, due to local eutrophication near the shoreline [56]. However, *O. flavus* was unable to effectively cope with live *E. verrucosus* under these conditions, highlighting the complex ecological differences between littoral and deep-water species.

## 5. Conclusions

Here, for the first time, we estimated the sensitivity of endemic amphipods from Lake Baikal to environmentally relevant levels of ultraviolet radiation and demonstrated reduced UV tolerance in the deep-water scavengers. The sensitivity to UV was even more pronounced for deeper-dwelling *O. albinus* than *O. flavus* and was partially explainable by the amounts of carotenoids and carotenoid-binding proteins in the amphipods. However, we could not determine the leading parameter among these two for UV protection. We found no signs of oxidative stress in the studied amphipods but UV-B specifically seemed to cause accumulation of some toxic compounds in two deep-water species. Unexpectedly, no elevated DNA damage was indicated in *O. albinus* under UV, when it was the case for *O. flavus*. Overall, the obtained results demonstrate that UV is an important factor limiting the distribution of deep-water amphipods into the littoral zone of Lake Baikal, but even a potential reduction in water transparency would be unlikely to allow them to compete with littoral species.

## Figures and Tables

**Figure 1 biology-13-01004-f001:**
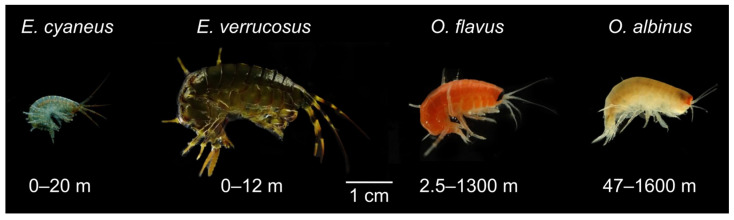
Photographs of adult endemic amphipods from Lake Baikal used in the study: littoral species *Eulimnogammarus cyaneus* (Dybowski, 1874) and *E. verrucosus* (Gerstfeldt, 1858) and deep-water species *Ommatogammarus flavus* (Dybowsky, 1874) and *O. albinus* (Dybowsky, 1874). Depth ranges for each species distribution are given according to [13,20,21]. Photo credits: Elizaveta Kondrateva and Polina Drozdova.

**Figure 2 biology-13-01004-f002:**
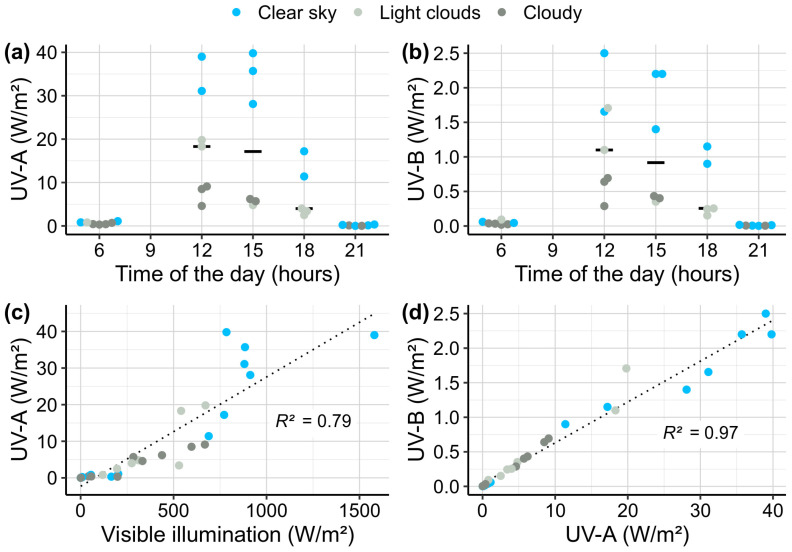
UV monitoring on the shoreline of Lake Baikal (Bolshie Koty village, 4–10 of July). (**a**) UV-A and (**b**) UV-B intensities corresponding to the 24 h time format. (**c**) Relation between visible solar illumination and UV-A intensity. (**d**) Relation between UV-A and UV-B intensities. Horizontal solid lines in (**a**,**b**) indicate medians. Dotted lines in (**c**,**d**) show linear regressions, and *R*^2^ indicates adjusted coefficient of determination for the models.

**Figure 3 biology-13-01004-f003:**
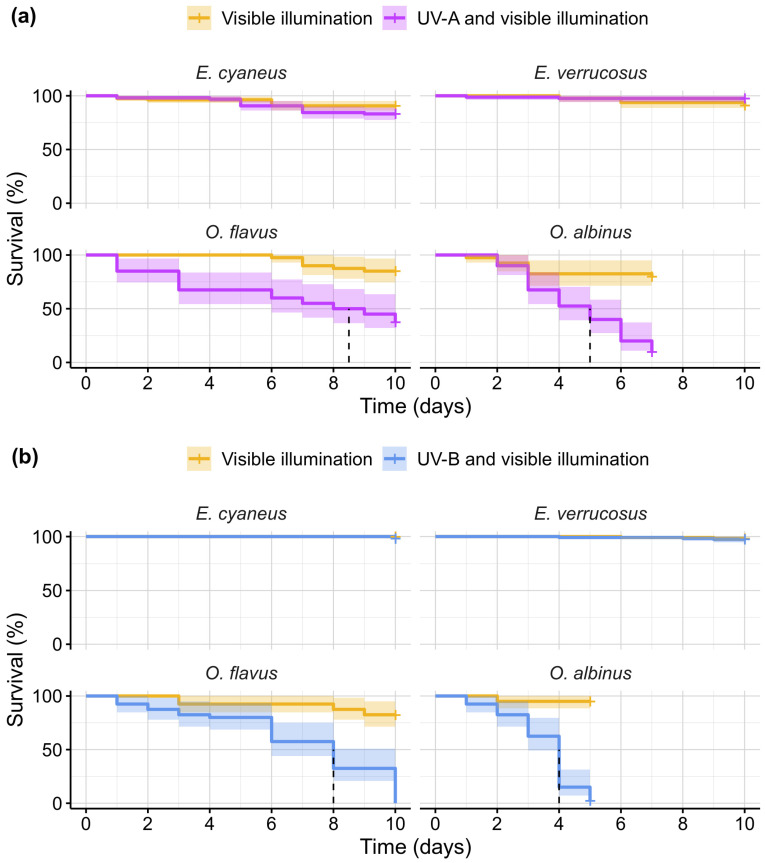
Survival rates of endemic amphipods from Lake Baikal (littoral species *E. cyaneus* and *E. verrucosus* and deep-water species *O. flavus* and *O. albinus*) during laboratory exposures under (**a**) UV-A and (**b**) UV-B treatments. Solid lines indicate the Kaplan–Meier survival curves, colored bands show the 95% confidence intervals for the curves and vertical dashed lines indicate reaching LT_50_. *n* = 160 for *E. cyaneus*; *n* = 80 (UV-A) or 120 (UV-B) for *E. verrucosus*; *n* = 40 for each *Ommatogammarus* species (see Appendix A for raw data).

**Figure 4 biology-13-01004-f004:**
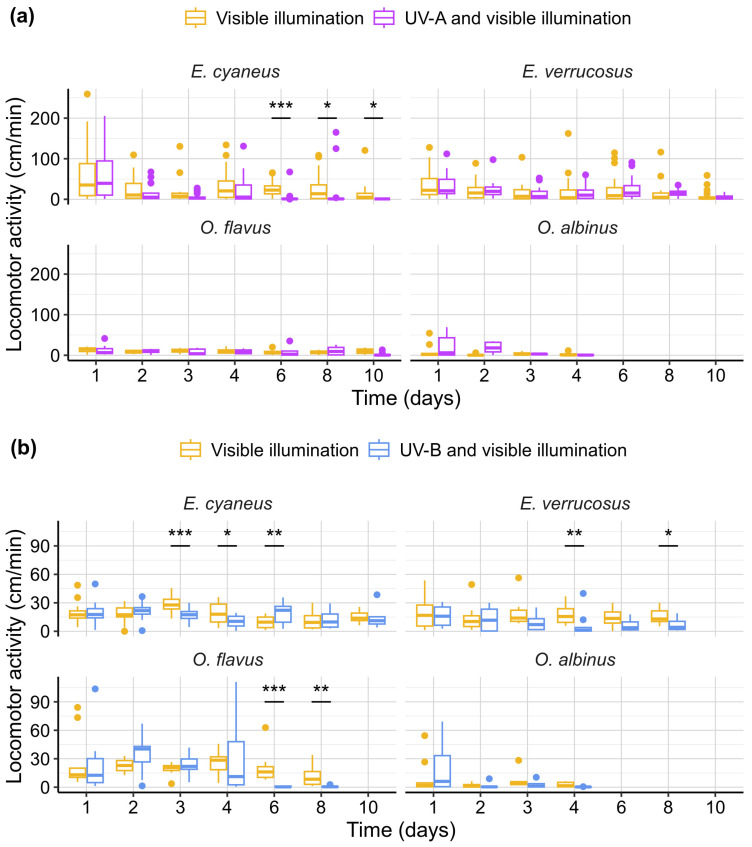
Horizontal locomotor activity of endemic amphipods from Lake Baikal (littoral species *E. cyaneus* and *E. verrucosus* and deep-water species *O. flavus* and *O. albinus*) during laboratory exposures under (**a**) UV-A and (**b**) UV-B treatments. *n* ≥ 8 for *Eulimnogammarus* species; *n* ≥ 4 for *Ommatogammarus* species (see Appendix A for raw data). * *p* < 0.05, ** *p* < 0.01, *** *p* < 0.001.

**Figure 5 biology-13-01004-f005:**
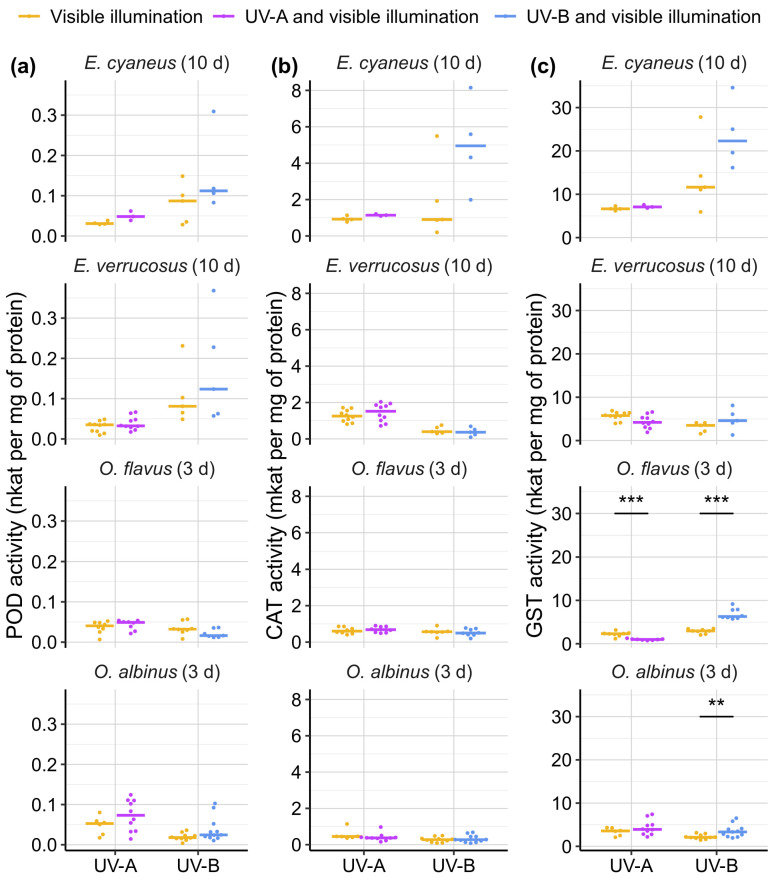
Activities of antioxidant enzymes (**a**) peroxidase (POD), (**b**) catalase (CAT), and (**c**) glutathione-S-transferase (GST) in endemic amphipods from Lake Baikal after laboratory exposures under UV-A and UV-B treatments. Note the difference in chosen exposure times: 10 days for littoral species (*E. cyaneus* and *E. verrucosus*) and 3 days for deep-water species (*O. flavus* and *O. albinus*). Horizontal colored solid lines indicate medians and dots show biological replicates. *n* ≥ 3 for *E. cyaneus*; *n* ≥ 5 for *E. verrucosus*; *n* ≥ 6 for *Ommatogammarus* species (see Appendix A for raw data). ** *p* < 0.01, *** *p* < 0.001.

**Figure 6 biology-13-01004-f006:**
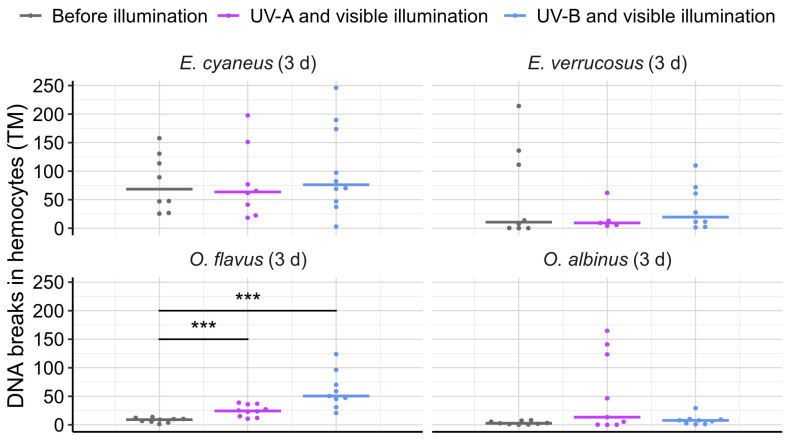
DNA breaks (expressed in tail moment, TM) in hemocytes of endemic amphipods from Lake Baikal (littoral species *E. cyaneus* and *E. verrucosus* and deep-water species *O. flavus* and *O. albinus*) after laboratory exposure under UV-A and UV-B treatments during 3 days. Horizontal colored solid lines indicate medians and dots show biological replicates. *n* ≥ 5 for *E. verrucosus*; *n* ≥ 8 for other species (see Appendix A for raw data). *** *p* < 0.001.

**Figure 7 biology-13-01004-f007:**
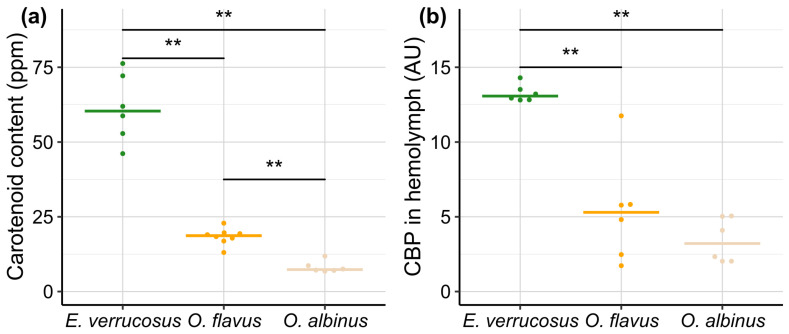
Parameters related to UV protection in endemic amphipods from Lake Baikal (littoral *E. verrucosus* and deep-water species *O. flavus* and *O. albinus*). (**a**) Total carotenoid content in the body (expressed in parts per million—ppm). (**b**) The level of a carotenoid-binding protein (CBP) in hemolymph (expressed in arbitrary units—AU). Note that the animals were kept in laboratory conditions for a short time without stressful exposures, used for hemolymph extraction and, finally, frozen for measurement of carotenoid content. Horizontal colored solid lines indicate medians and dots show biological replicates. *n* ≥ 6 for all species (see Appendix A for raw data). ** *p* < 0.01.

**Figure 8 biology-13-01004-f008:**
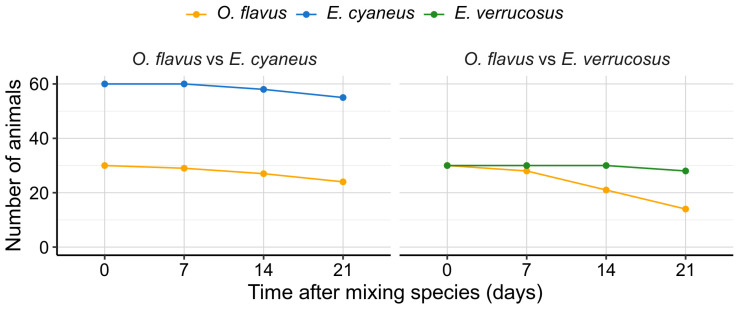
Survival rates of deep-water *O. flavus* kept either with littoral *E. cyaneus* or littoral *E. verrucosus* in two aquaria with large surfaces and with stones. UV radiation was excluded, while visible illumination was dim and followed the natural diurnal cycle. Dots show the results of counting amphipods in these two aquaria. See Appendix A for raw data.

## Data Availability

Raw obtained data are presented as the Appendix A to this article in Appendix A.

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
