# Peer review of "UV Sensitivities of Two Littoral and Two Deep-Freshwater Amphipods (Amphipoda, Crustacea) Reflect Their Preferred Depths in the Ancient Lake Baikal"

_biology, 2024, doi:10.3390/biology13121004_

Round 1

Reviewer 1 Report (Previous Reviewer 1)

Comments and Suggestions for Authors

In this version of the manuscript the authors have correctly interpreted the reviewers' suggestions. In relation to my comments, I was satisfied with the authors' explanations regarding the characteristics of the lamp used and the changes they made when eliminating UVC information.The change in the title was more appropriate to the research work and the efforts of the authors to improve Abstract, Math and Meth, epigraphs and Discussion, are recognized.

With all these changes made, I consider that the manuscript is now appropriate to be published in Biology.

Author Response

In this version of the manuscript the authors have correctly interpreted the reviewers' suggestions. In relation to my comments, I was satisfied with the authors' explanations regarding the characteristics of the lamp used and the changes they made when eliminating UVC information.The change in the title was more appropriate to the research work and the efforts of the authors to improve Abstract, Math and Meth, epigraphs and Discussion, are recognized.

With all these changes made, I consider that the manuscript is now appropriate to be published in Biology.

Authors: Dear Reviewer 1, thank you very much for critically reading our updated manuscript and for the high estimate of our work.

Reviewer 2 Report (Previous Reviewer 3)

Comments and Suggestions for Authors

The authors have done a good job of addressing my comments and concerns in the first round of review, and I have only a few minor considerations for the authors listed below. I look forward to seeing where this research goes, especially with the potential for measurement of UV-induced CPDs in this study system.  

Specific comments

Section 2.3: This section is still a bit difficult to comprehend and would benefit from further re-wording. One suggestion is that you could allocate names to the different ‘types of exposures’ as you refer to them (L161). Perhaps you can refer to them as the UV-A treatment and the UV-B treatment throughout the manuscript. Related to this, you mention each treatment is composed of multiple lamps, so are the measured irradiances specified in L171-177 specific to individual lamps, or are they the overall treatment irradiance  (that is, the overall irradiance produced by all the lamps within a treatment)? If the latter, which I suspect is the case, then I suggest that you re-word from ‘lamps’ to ‘treatment’, e.g., “5.2 W/m2 of UV-A and 0.16 W/m2 of UV-B for the UV-A treatment, and ~0.4 W/m2 of UV-B and ~1 W/m2 of UV-A for the UV-B treatment”. Same for L340-341.

Line180–184: I think it would be okay to shorten these sentences for the sake of clarity and simplicity, e.g.,: “Most experiments (except for the comet assay) consisted of four aquaria per UV treatment per species (each is considered an independent replicate), as well as four aquaria for the respective UV-A and UV-B control treatments with only visible illumination.”

Line 190-192: I am just checking the wording here. Were the littoral species tested sequentially (one after the other) or simultaneously (at the same time)? Same for the deep-water species.

L193–196: the sentence “the total number of experimental aquaria under UV-A was 8 for each littoral species”, makes it sounds like they were all under UV-A at the same time, which contradicts the statement earlier in the methods that each treatment was 4 aquaria (L180-184). I assume the authors are trying to refer to the total number of replicates run during the study (4 aquaria x 2 separate experiments = 8), in which case I would change the word aquaria to replicates: “the total number of UV-A treatment replicates was 8 for each littoral species”. Same for (L195). If my interpretation of the experimental design is incorrect, then I hope that the authors manage to re-word this section for better clarification.

L305–306: A short justification for reporting the p-values without correction for multiple comparison should be provided here, as the reader will not understand why this is the case.

L347-350: this sentence needs to be clarified because it is not clear what effect each X2 value is referring to. I suggest summarising like this: (all four comparisons: χ2 ≤ 3.7;  p ≥ 0.06). Same for L352-353, and L357. The alternative is to write the exact X2 and p value for each effect/comparison, e.g., (E. cyaneus: χ2 = 3.7;  p = 0.06 and χ2 = 3;  p = 0.12 for UVA and UVB treatments, respectively).

Well done to the authors for their efforts in refining this manuscript.

Author Response

The authors have done a good job of addressing my comments and concerns in the first round of review, and I have only a few minor considerations for the authors listed below. I look forward to seeing where this research goes, especially with the potential for measurement of UV-induced CPDs in this study system.  

Authors: Dear Reviewer 2, thank you very much for helping us to substantially improve the manuscript and for proof-reading the updated version! (Please check the attached file for more convenient reading).

Specific comments

Section 2.3: This section is still a bit difficult to comprehend and would benefit from further re-wording. One suggestion is that you could allocate names to the different ‘types of exposures’ as you refer to them (L161). Perhaps you can refer to them as the UV-A treatment and the UV-B treatment throughout the manuscript. Related to this, you mention each treatment is composed of multiple lamps, so are the measured irradiances specified in L171-177 specific to individual lamps, or are they the overall treatment irradiance  (that is, the overall irradiance produced by all the lamps within a treatment)? If the latter, which I suspect is the case, then I suggest that you re-word from ‘lamps’ to ‘treatment’, e.g., “5.2 W/m2 of UV-A and 0.16 W/m2 of UV-B for the UV-A treatment, and ~0.4 W/m2 of UV-B and ~1 W/m2 of UV-A for the UV-B treatment”. Same for L340-341.

Authors: Yes, the specified irradiance is the overall values (the latter), not individual lamps. Thus, we’ve switched to «UV treatment» instead of «UV lamps» throughout the text as suggested. We especially appreciate this valuable comment.

Line180–184: I think it would be okay to shorten these sentences for the sake of clarity and simplicity, e.g.,: “Most experiments (except for the comet assay) consisted of four aquaria per UV treatment per species (each is considered an independent replicate), as well as four aquaria for the respective UV-A and UV-B control treatments with only visible illumination.”

Authors: Corrected as suggested with minor rephrasing.

Line 190-192: I am just checking the wording here. Were the littoral species tested sequentially (one after the other) or simultaneously (at the same time)? Same for the deep-water species.

Authors: Yes, the wording is correct here as the experiments were done sequentially, i.e. one species after the other.

L193–196: the sentence “the total number of experimental aquaria under UV-A was 8 for each littoral species”, makes it sounds like they were all under UV-A at the same time, which contradicts the statement earlier in the methods that each treatment was 4 aquaria (L180-184). I assume the authors are trying to refer to the total number of replicates run during the study (4 aquaria x 2 separate experiments = 8), in which case I would change the word aquaria to replicates: “the total number of UV-A treatment replicates was 8 for each littoral species”. Same for (L195). If my interpretation of the experimental design is incorrect, then I hope that the authors manage to re-word this section for better clarification.

Authors: Yes, you are right as 4 aquaria x 2 separate experiments = 8. We’ve corrected as suggested.

L305–306: A short justification for reporting the p-values without correction for multiple comparison should be provided here, as the reader will not understand why this is the case.

Authors: We’ve added here the summarized motivation from our response during the first review round.

L347-350: this sentence needs to be clarified because it is not clear what effect each X2 value is referring to. I suggest summarising like this: (all four comparisons: χ2 ≤ 3.7;  p ≥ 0.06). Same for L352-353, and L357. The alternative is to write the exact X2 and p value for each effect/comparison, e.g., (E. cyaneus: χ2 = 3.7;  p = 0.06 and χ2 = 3;  p = 0.12 for UVA and UVB treatments, respectively).

Authors: We’ve summarized the values as suggested.

Well done to the authors for their efforts in refining this manuscript.

Authors: Thank you very much again for your work and the provided suggestions!

This manuscript is a resubmission of an earlier submission. The following is a list of the peer review reports and author responses from that submission.

Round 1

Reviewer 1 Report

Comments and Suggestions for Authors

The manuscript “Gradual loss of UV tolerance with increasing habitat depths in  deep-freshwater amphipods (Amphipoda, Crustacea) of ancient  Lake Baikal Follow a topic about to determine the effect of Ultraviolet radiation on some species of amphipods endemic to Lake Baikal  in relation to the depth of the lake and the presence of sunscreens in the animals . In this manuscript, correct sampling and a good presentation of the materials and methods by the authors are observed, but I consider that some mistakes have been made in the experimentation and I consider some shortcomings in the interpretation of the results and conclusions, so I do not consider that a relevant evaluation can be made. I provide some specific comments for the authors (by lines) below:

Lines 160/161. The lamps used in the experiment present UVC which should be eliminated with some filter since in nature UVC is absorbed by the ozone layer and does not reach the surface of the earth and is very harmful so it would be influencing the results,  unless the reason for using it is clarified in the results or discussion.

Line 164. Visible light should be in the same units as UV (W/m2).

Regarding the interpretation of results,  in my opinion the most important “bump”,  would be missing more in-depth analysis of UV absorbing compounds such as mycosporin like aminoacids * MAAs or any other compound that absorbs within the 280/320 wavelength (UV range ) or else comment on it in the Discussion

The conclusion from line 418 that UV radiation might be a contributing factor in limiting the vertical distribution of the deep-water Baikal amphipods, particularly of the more UV sensitive O. albinus. Perhaps it is somewhat obvious and not relevant given other works with albino amphipods in lakes.

The references are appropriate and manuscript citations are up to date, although I consider this is not the manuscript to be published and numerous changes, including experimental ones, must be made.

Author Response

Comments and Suggestions for Authors

The manuscript “Gradual loss of UV tolerance with increasing habitat depths in  deep-freshwater amphipods (Amphipoda, Crustacea) of ancient  Lake Baikal Follow a topic about to determine the effect of Ultraviolet radiation on some species of amphipods endemic to Lake Baikal  in relation to the depth of the lake and the presence of sunscreens in the animals . In this manuscript, correct sampling and a good presentation of the materials and methods by the authors are observed, but I consider that some mistakes have been made in the experimentation and I consider some shortcomings in the interpretation of the results and conclusions, so I do not consider that a relevant evaluation can be made. I provide some specific comments for the authors (by lines) below:

Authors: Dear Reviewer 1, thank you very much for critically reading the manuscript and high estimate of our presentation of the results. Please find our responses to the specific comments below.

Lines 160/161. The lamps used in the experiment present UVC which should be eliminated with some filter since in nature UVC is absorbed by the ozone layer and does not reach the surface of the earth and is very harmful so it would be influencing the results,  unless the reason for using it is clarified in the results or discussion.

Authors: Thank you very much for noticing these details as it really helped us to resolve this important issue in presenting the results! We of course knew that sun’s UV-C radiation reaching Earth’s surface is very low, but since all the lamps that were used for the experimental exposures were designed specifically for sun-mimicking illumination of human and animal skin (the UV-A lamps are actually common tanning lamps, and the UV-B lamps are intended for terrariums with desert animals), we didn’t check the specific values in other studies. As we see now in the recent literature, the sun UV-C levels below the atmosphere are indeed barely detectable even with modern sensitive spectrometers, while the used UV-radiometer showed us measurable results for UV-C on the shoreline of Lake Baikal (data were presented in original Table S1).

With this discrepancy in mind, we carefully checked again the manual for the used UV-radiometer. It is a governmentally certified device with high sensitivity, so we didn’t question its technical details previously. We’ve figured out that specifically for UV-C this UV-radiometer is intended for measurements only from germicidal UV-C lamps. We also checked the spectral ranges for three filters of the device in the manual and found that the UV-C channel is partially sensitive to UV-B as well (in the range of 280–290 nm), since the radiometer is just not designed for UV-C measurements from the sources with prevailing UV-B output, like the sun. It completely explains why we observed the "UV-C" readings both from the sun and the used lamps. Importantly, the "UV-C" readings from the lamps were over seven-fold lower than the median values from the sun, additionally supporting that the readings are actually due to UV-B.

Thus, since specifically the UV-C measurements with this UV-radiometer are unreliable for our sources and we don’t have any indications that the used UV lamps have UV-C output, we removed all information about the UV-C measurements from the manuscript.

Line 164. Visible light should be in the same units as UV (W/m2).

Authors: Thank you for this comment! We’ve indeed figured out how to convert illuminance to irradiance for the solar light, so the values for visible light are now presented in W/m2 in Figure 2 as suggested in order to make the numbers more understandable for the readers. However, the spectra of the used lamps are different from the solar spectrum, and in their case the process of conversion to irradiance would be different. Moreover, the main point in the measurements of the visible light intensities for the lamps was to equilibrate them for amphipod eyes. Since illuminance is more suitable than irradiance specifically for this purpose, we’ve decided to keep these values for the lamps.

Regarding the interpretation of results,  in my opinion the most important “bump”,  would be missing more in-depth analysis of UV absorbing compounds such as mycosporin like aminoacids * MAAs or any other compound that absorbs within the 280/320 wavelength (UV range ) or else comment on it in the Discussion

Authors: This is indeed an important remark, thank you. As we don’t have enough samples left right now (the deep-water species are quite laborious to collect and the process is not possible in all seasons), we’ve acknowledged this drawback in the most relevant sentence in the Discussion as you suggested: “Although here we could not estimate other important UV-screening compounds such as mycosporines and mycosporine-like amino acids, these findings are consistent with the well-established role of carotenoids as protective pigments against UV radiation”. We also softened any conclusions regarding carotenoids as they only partially explain the observed results.

The conclusion from line 418 that UV radiation might be a contributing factor in limiting the vertical distribution of the deep-water Baikal amphipods, particularly of the more UV sensitive O. albinus. Perhaps it is somewhat obvious and not relevant given other works with albino amphipods in lakes.

Authors: It’s now clear to us that we hadn’t given enough literature context along the Discussion; we apologize for that. The Reviewer 3 had a somewhat similar comment and requested more links to previous works specifically in the Discussion. Now we’ve significantly expanded the part of Discussion placing our results on amphipod survival in the modern context. The studies describing UV effects on aquatic organisms are numerous, and we tried to highlight only the relevant examples. We (once again) spent quite a lot of time checking the literature, but, surprisingly, didn’t find such examples specifically for deep-sea amphipods. Additionally, since Lake Baikal is the only freshwater reservoir with true and constantly oxygenated deep-water zone (which is not the case for Titicaca, Tanganyika or smaller lakes), such examples from other lakes are practically absent. There is, of course, a multitude of pale amphipods in caves and related freshwater ecosystems such as Gammarus minus, but we could find only very preliminary remarks on their UV sensitivity without presented data. Nevertheless, we still decided to acknowledge such an interesting model in Discussion. We believe that this context additionally demonstrates the importance of deep-water scavengers from Lake Baikal as a convenient study system.

The references are appropriate and manuscript citations are up to date, although I consider this is not the manuscript to be published and numerous changes, including experimental ones, must be made.

Authors: Thank you very much again for your work and the provided suggestions.

Reviewer 2 Report

Comments and Suggestions for Authors

This is a very interesting and well planned out research project and paper. My only that you mention how climate change can impact UV penetration in water bodies, but you don't explain how climate change might subject deeper water amphipods to clearer water and more UV rays. If you are making the point that the deeper amphipods might move to shallower habitats, please emphasize this a bit more. I think the project is great and has merit, I would just like to see more description of the significance of the research as we already know that UV light will impact range of organisms.   

Author Response

Comments and Suggestions for Authors

This is a very interesting and well planned out research project and paper. My only that you mention how climate change can impact UV penetration in water bodies, but you don't explain how climate change might subject deeper water amphipods to clearer water and more UV rays. If you are making the point that the deeper amphipods might move to shallower habitats, please emphasize this a bit more. I think the project is great and has merit, I would just like to see more description of the significance of the research as we already know that UV light will impact range of organisms.   

Authors: Dear Reviewer 2, thank you very much for evaluating the manuscript and this relevant suggestion. Indeed, the initial ecological idea behind comparing littoral and deep-water species was to check if Ommatogammarus scavengers are able to invade the sub-littoral or even littoral zone of Lake Baikal and compete with Eulimnogammarus species if UV levels decrease due to local eutrophication caused by human activities (mentioned at the end of Discussion) or more global processes such as climate change. Since the experiment presented in Fig. 8 clearly showed such outcome is hardly possible (in fact, originally we expected Ommatogammarus to have much higher predatory potential), we didn’t highlight the idea in Introduction and only briefly explained it at the very end of Discussion. From your comment we now see that it was a mistake, and in the updated manuscript we’ve added the following final sentence to the paragraph introducing Lake Baikal: «Besides evolutionary significance, UV intolerance of deep-water animals might have ecological implications for predicting their migration patterns if the lake water becomes less transparent for solar UV». We would not like, however, to further speculate on this possibility since littoral species are clearly also a factor limiting distribution of O. flavus even without the pressure of solar UV.

Authors: Thank you very much again for your work and the provided suggestion.

Reviewer 3 Report

Comments and Suggestions for Authors

Summary

The aim of this study was to assess the UV sensitivity of four amphipod species  that preferentially occupy different depths of Lake Baikal. The authors hypothesised that UV sensitivity correlates with preferred habitat depth, reflecting a gradual loss of energetically costly UV protective mechanisms in deep-water evolved species. To test this hypothesis, the authors exposed wild-caught amphipods to various treatments of artificial UVA, UVB and visible lighting under controlled laboratory conditions. The results generally supported the hypothesis, with multi-day UV exposure causing greater mortality in deep-water species compared to littoral species. Furthermore, deep-water species had lower carotenoid content (a UV-screening compound) in their hemolymph, which was likely related to the increased sensitivity to UV. The authors also measured locomotor activity, antioxidant activity and DNA damage, however issues with UV treatments fading in intensity, and the impact of mortality on sample sizes, meant that these data were not easily interpretable.

General comments

In my opinion, the study makes a significant and worthwhile contribution to the literature by providing evidence for reductive evolution using an intriguing and unique study system and demonstrating how UV radiation can be a limiting factor in the distribution of aquatic organisms throughout the water column. That said, I have a few significant concerns regarding the methods section that would need to be addressed prior to publication (see below). I have also provided several specific comments that I hope will help to further improve the manuscript.

1. The experimental design appears mostly sound, and control treatments are in place. However, important pieces of information that the reader needs to replicate this study are missing or are described in the results section instead of the methods. As such, it is currently unclear from the methods section how many experiments were conducted, what their duration was, and what the response variables and treatment samples sizes were in each experiment. This information needs to be explicitly stated in a structured manner in the experimental design section. On a related note, the results section should not repeat or state information pertaining to how the study was conducted – this information needs to go in the methods section (see specific comments for suggested changes).

2. The section “2.9. Data analysis and statistics” (L246-255) needs to provide more details regarding the description of statistical methods. Please specify the models and functions used to analyse the data, and whether a Mann-Whitney test was used for all of the response variables or just some. I also have some questions regarding the choice of statistics. Why was a Mann Whitney test used instead of the more powerful t-test? I am guessing some assumptions of the t-test prevented its used, but this should be mentioned in text. Why was a container effect not considered (given that animals were not housed individually)? The authors specify that UV sensitivity may be size-specific; could size be included as a covariate for the enzyme for the assay results? On a related note, it is standard practise to include more information to accompany p-values when describing statistical outputs in the Results section. In the case of the Mann-Whitney test, I believe it is a U test statistic or z value. Please update results with additional statistics to accompany the p values.  

3. I am concerned with the use of the standard comet assay for detection of UV-induced DNA damage. This is because the primary DNA damage caused by UV-B radiation is pyrimidine dimers (particularly CPDs), which are silent lesions that are not detected by the standard comet assay. Pyrimidine dimers are covalent bonds between adjacent pyrimidine bases that cause kinks in the DNA which interfere with transcription and replication, but they are not strand breaks and so do not migrate in the comet assay gel. In order to detect CPDs in addition to single and double strand breaks, an Enzyme-Modified Comet Assay utilising T4 endonuclease V is needed to induce single-strand breaks at the CPD sites, prior to running gel electrophoresis (some relevant papers are referenced below). If it is not possible to re-run these assays, the limitations of the standard comet assay at least need to be discussed in the discussion, including some thoughts on the implications of not detecting UV-induced CPDs on the results of the study. This represents an important area of further study.

Peter Møller, The comet assay: ready for 30 more years, Mutagenesis, Volume 33, Issue 1, January 2018, Pages 1–7, https://doi.org/10.1093/mutage/gex046

Muthusamy, Ganesan; Balupillai, Agilan; Govindasamy, Kanimozhi; Ramasamy, Karthikeyan; Ponniresan, Veeramani Kandan; Malla, Illiyas Magbool; Nagarajan, Rajendra Prasad. Modified Comet Assays for the Detection of Cyclobutane Pyrimidine Dimers and Oxidative Base Damages. Journal of Radiation and Cancer Research 8(1):p 82-86, Jan–Mar 2017. | DOI: 10.4103/0973-0168.199312

4. Discussion (L386-486): The discussion is generally well written but is lacking comparisons with other studies in similar systems (e.g., the oceans). Comparing results to other studies helps to place the findings into context. On a separate note, figures that are cited in the results are generally not referred to again in the Discussion section (the reader is already aware of all the figures at this point).

Specific comments

Title: the term “gradual loss”  is misleading because it has a double meaning and gives the impression that the study investigated a temporal component of UV tolerance – either from an evolutionary point of view, or from an acclimation point of view (within the lifetime of individual organisms as they shift through the water column). Given that the study simply tested species-specific UV tolerances (and not speciation/evolution), I recommend the title be amended to better reflect the findings of the study, e.g., “UV sensitivity of four freshwater amphipods (Amphipoda, Crustacea) reflect preferred depths in ancient Lake Baikal”. Alternatively, if the authors wish to retain title, I suggest changing to the word ‘evolutionary’ to imply long-term evolutionary changes as opposed to acclimation: “Evolutionary loss of UV tolerance…”. Additionally, the Intro and Discussion section should take on a more evolution-focussed angle in this case, including more discussion of the history of the formation of the lake and the descendance of the deepwater amphipod species from the littoral species.

Abstract (L26-29): The phrase “recently descending” is confusing because it has a double meaning – either recently evolved from (which is a temporal phenomenon), or moving to a greater depth of the water column (which is a spatial phenomenon). Please clarify this sentence.

Abstract (L32): a short description of the methods is necessary for the reader to understand the context of the work and findings (that is, a laboratory setting with artificial lighting as opposed to a field setting in Lake Baikal). This should include some indication of the duration of the UV exposure.

Abstract (L34-36): this statement is overreaching as the study did not decipher all the potential mechanisms of UV tolerance. Correct to “partially explainable”, or “related to” or similar. Same for L490-492.

L107-108: good introduction, but this statement has no context and could use some fleshing out. Namely, what was the point of this experiment? Was it to determine if UV may be acting as  the sole factor limiting the distribution of deep-water amphipods (as opposed to competition and predation etc.)?

Materials and Methods structure: The first section feels out of place and is missing experimental context. I suggest starting with section 2.2, whilst moving L111-116 after the UV treatments have been described in section 2.3. L117-121 could be moved to the start of Section 2.3, describing how they were used to place the experimental UV levels into an ecologically relevant context.

L113 and L114: are the specific wavelength ranges for the UVA, UVB and UVC sensors (and the combined UVA-UVB sensor) known? If so, please specify in the text, as they can vary slightly between instruments.

L119: I assume the authors are referring to 12 pm in the middle of the day, not 12 am. Correct also on L262 as well.

L135: Is there a reason why the littoral and deep-water species were collected at different times of the year? Does this mean that experiments on these species were conducted at different times? This type of information needs to be specified in the methods section because it has implications for results and repeatability of the work.

L149: Section 2.3 switches between present and past tense. Please update so that the methods section is all past tense.

L159-167: This level of decline in UV outputs is surprising given that exposures were maximum 10 days. Reasons for this decline, and recommendations for avoiding this issue in future work should be provided in the discussion.

L174: Please specify why no stones were provided – was it to prevent shading opportunities?

L180: Please clarify what ‘see below’ is referring to.

L184-185: More details are needed for the estimation of locomotor activity. What were the time points? Were amphipod movements tracked individually? Which containers were they in?

L187: so only 1 min out of the 5 min of recording was used? Was the minute chosen randomly selected?

L187: there are multiple ways to measure movement (stop-start activity, acceleration etc.). Please specify the type of movement that was measured, e.g., distance over time (cm/min).

L190: How long was the UV exposure period? How many is ‘a group’?

L208-212: some explanation of the unit (tail moment) is warranted given that this is not assumed knowledge for the generalist readership of biology journal.

Section 3.1 (L257-284): were experimental UV treatments designed after considering these environmental measurements? If this is the case, then the authors have done a fine job in attempting to maximise the ecological relevance of their experiment. However, it seems odd to describe these environmental measurements after the experimental levels and not before. In my opinion (this is optional), this section would make more sense at L149, before the experimental treatments are described. That way the reader is more aware of how the experimental UV conditions were chosen.

L260, L263: when referring to UVA and UVB levels (W/m2), the correct term is irradiance, not illuminance.

L267-L268: I am unsure of how UVC was detected in the field given that it is fully absorbed by the Earth’s atmosphere. Furthermore, I am unsure of the ecological relevance of the statement and I would consider removing it.

L278-279: Was this percentage including or excluding the effect of the glass on the sensor?  Please clarify in text.

L279: Table S1 should be clearly labelled as such within the supplementary file.

L286-289: methods should not be repeated in the results section, I suggest deleting. Same here: L327-328.

L292: this is the first mention of the duration of the experiment, which should be stated in the experimental design section of the methods.

L300: Have the authors considered a statistical comparison between UV-A and UV-B treatments for each species? Or statistical comparison between deepwater species? This could provide statistical evidence to the claims that O. albinus is more sensitive than O. flavus.

L305-307: This information needs to be provided in the methods instead. Same for other parts of the results section: L328-331, L342-344, L352-355, L372-377.

L310: please clarify in text what comparisons the two p values are referring to.

L382-385: Were control treatments conducted (each species living separately under the same conditions)? Without this control, it is not possible to determine whether co-inhabitation of O. flavus and E. cyaneus affected their survival, or if it was just background mortality.

L449-451: To confirm the theory that the treatments did not induce significant ROS production would require the direct measurement of ROS or the damage it causes (e.g., protein carbonyl and lipid peroxidation). These future directions should be recognised in text.

L454-L465: These are all plausible explanations, but it is important to acknowledge in the discussion that this assay did not measure the primary type of DNA damage caused by UV radiation.

L498-500: this last sentence is confusing. Maybe change “would not probably” to “would be unlikely to”.

L515: I am not sure what Table S1 is referring to.

Figures

Figure 2 could probably go into supplementary material.

Some general tips for tidying all the figures include (1) placing units in brackets instead of separating with a comma and (2) removing gridlines. Figure panels should be labelled (a), (b) and so on to make it easier to refer the reader to specific graphs within the figures. Figure legends also need to define whether horizontal lines in some figures represent mean or median.

Figure 2: it may be worthwhile including a line of best fit on panels C and D. There is an unusual ‘clear sky’ point with very low irradiance at 15:00. Consider whether there is a reasonable explanation for this value or whether it should be considered a potential error/outlier for removal?

Figure 2 caption (L270-273): This figure caption is repetitive and can be simplified, e.g., UV-A (a) and UV-B (b) intensities corresponding to the 24-hour time format.

Figure 3: Please specify in the figure caption what the coloured shading around the solid survival trend lines indicate.

Figure 4: There are some major points of unclarity with this figure and the locomotor performance data. Firstly, it is not clear what samples sizes are being worked with, and whether mortality is affecting this. Presumably, the lack of data for O. albinus after day 6 is due to mortality. Sample size ranges should be provided in figure captions or described in the methods. Secondly, the x-axis is very random because it misses day 4 and 5, and then jumps to a 2-day interval. Why were some days excluded from the figure?

Figure 5, 6 and 7: Some form of error also needs to be shown (e.g., standard error, standard deviation, confidence interval etc.).

Figure 8: This time-series data is better presented as a line graph

Author Response

Comments and Suggestions for Authors

Summary

The aim of this study was to assess the UV sensitivity of four amphipod species  that preferentially occupy different depths of Lake Baikal. The authors hypothesised that UV sensitivity correlates with preferred habitat depth, reflecting a gradual loss of energetically costly UV protective mechanisms in deep-water evolved species. To test this hypothesis, the authors exposed wild-caught amphipods to various treatments of artificial UVA, UVB and visible lighting under controlled laboratory conditions. The results generally supported the hypothesis, with multi-day UV exposure causing greater mortality in deep-water species compared to littoral species. Furthermore, deep-water species had lower carotenoid content (a UV-screening compound) in their hemolymph, which was likely related to the increased sensitivity to UV. The authors also measured locomotor activity, antioxidant activity and DNA damage, however issues with UV treatments fading in intensity, and the impact of mortality on sample sizes, meant that these data were not easily interpretable.

General comments

In my opinion, the study makes a significant and worthwhile contribution to the literature by providing evidence for reductive evolution using an intriguing and unique study system and demonstrating how UV radiation can be a limiting factor in the distribution of aquatic organisms throughout the water column. That said, I have a few significant concerns regarding the methods section that would need to be addressed prior to publication (see below). I have also provided several specific comments that I hope will help to further improve the manuscript.

Authors: Dear Reviewer 3, thank you very much for critically reading the manuscript and high estimate of our research. Please find our responses to the specific comments below (there is also an attached file with this response that might be more convenient to read).

  1. The experimental design appears mostly sound, and control treatments are in place. However, important pieces of information that the reader needs to replicate this study are missing or are described in the results section instead of the methods. As such, it is currently unclear from the methods section how many experiments were conducted, what their duration was, and what the response variables and treatment samples sizes were in each experiment. This information needs to be explicitly stated in a structured manner in the experimental design section.

Authors: The ending of section 2.3 describing the experimental design is now significantly expanded in order to fully meet this request. Additionally, the numbers of individuals used per sample are currently added to the sections describing the enzymatic and comet assays. In part, we previously didn’t spell out the specific sample sizes in figure captions and Methods since we included the Table S1 with all the raw data into the submission (more comments on Table S1 below in respective responses); it’s also corrected of course.

On a related note, the results section should not repeat or state information pertaining to how the study was conducted – this information needs to go in the methods section (see specific comments for suggested changes).

Authors: We totally agree that the Methods section must contain all the technical details necessary to replicate the research and tried to be as clear as possible in the updated version of Methods. We also agree that some deep technical details in the previous version of Results were excessive and should have been moved to Methods (please see specific responses below). However, a lot of readers now do not check the Methods at all and directly go to the Results. Thus, we believe it’s not just optional but mandatory to briefly repeat some key points of the experimental design that are important to understand the details of the obtained data and also the motivation for specific analyses in certain cases in order to keep the reader in the context. Checking six most recent papers on https://www.mdpi.com/journal/biology showed us that at least three of them followed the same pattern and sometimes explained in the Results even deeper technical details than we do:

https://www.mdpi.com/2079-7737/13/11/932

https://www.mdpi.com/2079-7737/13/11/931

https://www.mdpi.com/2079-7737/13/11/929

Thus, we now moved the excessive deep technical details to Methods as suggested but retained some explanatory remarks in Results to keep the section understandable by itself.

  1. The section “2.9. Data analysis and statistics” (L246-255) needs to provide more details regarding the description of statistical methods. Please specify the models and functions used to analyse the data, and whether a Mann-Whitney test was used for all of the response variables or just some.

Authors: Yes, the Mann-Whitney test was used for comparisons of all measured parameters. This section is now expanded in order to provide more details as requested.

I also have some questions regarding the choice of statistics. Why was a Mann Whitney test used instead of the more powerful t-test? I am guessing some assumptions of the t-test prevented its used, but this should be mentioned in text.

Authors: The short answer is yes; since our the sample sizes were relatively low we could not test the data for normality and other basic assumptions for the t-test and its modifications for multiple comparisons, and this fact is now acknowledged in the text as requested. Diving into more details, the bare minimum for the sample size recommended for normality testing is 15 (Grech and Calleja, 2018, DOI: 10.1016/j.earlhumdev.2018.04.014) with many textbooks and statisticians suggesting at least 30 measurements (Fay and Gerow, 2013, DOI: 10.1895/wormbook.1.159.1), which we agree with. Most of the measured parameters included ~10 samples per group or less, and the only parameter where the sample size reached 20 samples was the locomotor activity. Thus, all the parameters were analyzed with the non-parametric test with the correction for multiple comparisons according to standard recommendations.

Why was a container effect not considered (given that animals were not housed individually)? The authors specify that UV sensitivity may be size-specific; could size be included as a covariate for the enzyme for the assay results?

Authors: Since each aquarium for amphipods usually requires aeration, it is practically infeasible to house each animal individually considering their numbers. We didn’t take into account the container effect since animals were chosen randomly from four aquaria for the enzymatic measurements. Now when you’ve mentioned it, we of course agree that it would be reasonable, but we didn’t record from which aquarium each individual was taken, unfortunately. We do have the information about weight of all samples used for the enzymatic measurements but for E. cyaneus, O. flavus and O. albinus many or all samples included over one individual due to their smaller size. Thus, such size-specific influence can be tested only for E. verrucosus but, since in this study we concentrate on comparing different species, it cannot gain any conclusions. However, we of course added to Methods the specific information about the number of individuals used per sample in different analyses to be clear.

On a related note, it is standard practise to include more information to accompany p-values when describing statistical outputs in the Results section. In the case of the Mann-Whitney test, I believe it is a U test statistic or z value. Please update results with additional statistics to accompany the p values.  

Authors: The U values are added. Additionally, we’ve now added χ2 to the survival results.

  1. I am concerned with the use of the standard comet assay for detection of UV-induced DNA damage. This is because the primary DNA damage caused by UV-B radiation is pyrimidine dimers (particularly CPDs), which are silent lesions that are not detected by the standard comet assay. Pyrimidine dimers are covalent bonds between adjacent pyrimidine bases that cause kinks in the DNA which interfere with transcription and replication, but they are not strand breaks and so do not migrate in the comet assay gel. In order to detect CPDs in addition to single and double strand breaks, an Enzyme-Modified Comet Assay utilising T4 endonuclease V is needed to induce single-strand breaks at the CPD sites, prior to running gel electrophoresis (some relevant papers are referenced below). If it is not possible to re-run these assays, the limitations of the standard comet assay at least need to be discussed in the discussion, including some thoughts on the implications of not detecting UV-induced CPDs on the results of the study. This represents an important area of further study.

Peter Møller, The comet assay: ready for 30 more years, Mutagenesis, Volume 33, Issue 1, January 2018, Pages 1–7, https://doi.org/10.1093/mutage/gex046

Muthusamy, Ganesan; Balupillai, Agilan; Govindasamy, Kanimozhi; Ramasamy, Karthikeyan; Ponniresan, Veeramani Kandan; Malla, Illiyas Magbool; Nagarajan, Rajendra Prasad. Modified Comet Assays for the Detection of Cyclobutane Pyrimidine Dimers and Oxidative Base Damages. Journal of Radiation and Cancer Research 8(1):p 82-86, Jan–Mar 2017. | DOI: 10.4103/0973-0168.199312

Authors: We completely agree with this criticism and now tried to acknowledge this limitation in the Discussion as fully as possible with the following sentences: «In the comet assay experiments, all species were exposed to the same dose of UV radiation but the used procedure did not include DNA pre-treatment with any endonucleases before the assay. Thus, we could not detect cyclobutane pyrimidine dimers that mostly appear under UV and could only find double- and single-stranded breaks» ... «However, the enzyme-modified comet assay should shed some more light on this topic».

We’ve recently received several endonucleases and are currently in the process of establishing the modified procedure in our lab but it is not ready yet. More importantly, the comet assay is performed on freshly extracted amphipod hemocytes, and freezing the samples is not an option (we tried it previously). Thus, we do not have the samples right now, and would not be able to obtain enough deep-water animals for the experiments until the beginning of spring due to seasonal conditions (solid ice is established on Lake Baikal at the end of February and sampling from a ship is hardly possible during winter). However, we are grateful for this remark and will follow the advice in future projects.

  1. Discussion (L386-486): The discussion is generally well written but is lacking comparisons with other studies in similar systems (e.g., the oceans). Comparing results to other studies helps to place the findings into context.

Authors: In the updated manuscript we’ve added the substantial part of Discussion placing our results on amphipod survival in the modern context. The studies describing UV effects on aquatic organisms are numerous, and here we tried to highlight only the relatable examples in the matter of object sizes and experimental approach. We (once again) spent quite a lot of time checking the literature, but, surprisingly, didn’t find such examples specifically on deep-sea amphipods, which may additionally highlight the importance of deep-water scavengers from Lake Baikal as a convenient study system.

On a separate note, figures that are cited in the results are generally not referred to again in the Discussion section (the reader is already aware of all the figures at this point).

Authors: Removed as suggested.

Specific comments

Title: the term “gradual loss”  is misleading because it has a double meaning and gives the impression that the study investigated a temporal component of UV tolerance – either from an evolutionary point of view, or from an acclimation point of view (within the lifetime of individual organisms as they shift through the water column). Given that the study simply tested species-specific UV tolerances (and not speciation/evolution), I recommend the title be amended to better reflect the findings of the study, e.g., “UV sensitivity of four freshwater amphipods (Amphipoda, Crustacea) reflect preferred depths in ancient Lake Baikal”. Alternatively, if the authors wish to retain title, I suggest changing to the word ‘evolutionary’ to imply long-term evolutionary changes as opposed to acclimation: “Evolutionary loss of UV tolerance…”. Additionally, the Intro and Discussion section should take on a more evolution-focussed angle in this case, including more discussion of the history of the formation of the lake and the descendance of the deepwater amphipod species from the littoral species.

Authors: Thank you for this thorough suggestion! The title is now changed to «UV sensitivities of two littoral and two deep-freshwater amphipods (Amphipoda, Crustacea) reflect their preferred depths in ancient Lake Baikal». We believe it’s important to mention here that two studied species are deep-water. Amphipod evolution within Lake Baikal is a complicated subject (in fact, it demands a separate research) with poor modern understanding of its links to the geological history; thus, we feel obliged to concentrate on the actual observed results rather than speculate on evolutionary details in addition to the already mentioned conceptions.

Abstract (L26-29): The phrase “recently descending” is confusing because it has a double meaning – either recently evolved from (which is a temporal phenomenon), or moving to a greater depth of the water column (which is a spatial phenomenon). Please clarify this sentence.

Authors: Corrected to «recently evolving»; thank you.

Abstract (L32): a short description of the methods is necessary for the reader to understand the context of the work and findings (that is, a laboratory setting with artificial lighting as opposed to a field setting in Lake Baikal). This should include some indication of the duration of the UV exposure.

Authors: Added as suggested. We now exactly reached the word limit for Abstract.

Abstract (L34-36): this statement is overreaching as the study did not decipher all the potential mechanisms of UV tolerance. Correct to “partially explainable”, or “related to” or similar. Same for L490-492.

Authors: Indeed, this is better phrasing. Now corrected in both places.

L107-108: good introduction, but this statement has no context and could use some fleshing out. Namely, what was the point of this experiment? Was it to determine if UV may be acting as  the sole factor limiting the distribution of deep-water amphipods (as opposed to competition and predation etc.)?

Authors: We re-wrote the sentence in the following way: «Additionally, we tested the ability of the deep-water species to prey on the littoral amphipods without the pressure of solar UV radiation to evaluate their potential to invade more shallow-water communities if the lake water becomes less transparent».

Moreover, the new sentence two paragraphs earlier also should give additional context: «Besides evolutionary significance, UV intolerance of deep-water animals might have ecological implications for predicting their migration patterns if the lake water becomes less transparent for solar UV».

Materials and Methods structure: The first section feels out of place and is missing experimental context. I suggest starting with section 2.2, whilst moving L111-116 after the UV treatments have been described in section 2.3. L117-121 could be moved to the start of Section 2.3, describing how they were used to place the experimental UV levels into an ecologically relevant context.

Authors: After a careful consideration we decided to treat the data from solar UV monitoring as part of the Results and not Methods (please see the response to the comment about section 3.1 below). As so, such a section in Results requires respective description in Methods, and the section 2.1 serves exactly this purpose. Additionally, it seems reasonable to first describe how we make the UV and visible light measurements throughout the study and next describe the experimental design with the specific UV/visible light levels measured with the earlier described tools.

L113 and L114: are the specific wavelength ranges for the UVA, UVB and UVC sensors (and the combined UVA-UVB sensor) known? If so, please specify in the text, as they can vary slightly between instruments.

Authors: The information is now added. To be as specific as possible, we mentioned not only the main sensitivity ranges for UV-A and UV-B channels but also the ranges with residual probe sensitivity (specified according to the spectra from manual to the main radiometer).

L119: I assume the authors are referring to 12 pm in the middle of the day, not 12 am. Correct also on L262 as well.

Authors: It’s now corrected, thank you!

L135: Is there a reason why the littoral and deep-water species were collected at different times of the year? Does this mean that experiments on these species were conducted at different times? This type of information needs to be specified in the methods section because it has implications for results and repeatability of the work.

Authors: Yes, since we didn’t keep the animals for a while before the exposures (the specific upper limits of acclimation are now also added), the experiments were conducted separately on littoral and deep-water amphipods in the seasons of their respective samplings. The difference in sampling times is purely due to seasonal features of Lake Baikal: littoral amphipods are most convenient to collect during summer and deep-water species are best collected from the ice cover at the very beginning of spring (among other factors, control of the sampling depth is much easier at this period). We now acknowledge these facts in the Methods as requested: «The difference in sampling times for littoral and deep-water amphipods is due to seasonal conditions on Lake Baikal and forced us to perform the most of further analyses on two pairs of species at different times».

L149: Section 2.3 switches between present and past tense. Please update so that the methods section is all past tense.

Authors: It’s now corrected.

L159-167: This level of decline in UV outputs is surprising given that exposures were maximum 10 days. Reasons for this decline, and recommendations for avoiding this issue in future work should be provided in the discussion.

Authors: The sets of lamps were used not only for this research but also for two additional studies (we are currently in preparation of a separate paper on the role of carotenoids and carotenoid-binding proteins in UV-protection on a convenient amphipod model from Baikal and also found UV intolerance for juveniles of the littoral species in a series of exposures). Some experiments also failed due to various reasons (such as power failure, temperature rises etc). The requested remarks are now added at the end of Discussion: “A notable drawback of the study was the decline in the lamp intensities, which was due to extensive use of the setup without the regular checks of its parameters. As so, use of such lamps would usually require exposing groups of different species strictly in parallel to allow direct comparison, while for the species-oriented experimental design as presented here more stable LED sources seem to be more appropriate”.

L174: Please specify why no stones were provided – was it to prevent shading opportunities?

Authors: It was exactly the motivation; it’s now added.

L180: Please clarify what ‘see below’ is referring to.

Authors: This phrase refers to the section describing how statistically significant differences between the survival curves were assessed. Now it is changed to: «see section 2.9 for the procedure description».

L184-185: More details are needed for the estimation of locomotor activity. What were the time points? Were amphipod movements tracked individually? Which containers were they in?

Authors: The parameters of aquaria are now mentioned in section 2.2. Yes, animal movements were tracked individually, and this information along with the time points are now added to section 2.4 as requested.

L187: so only 1 min out of the 5 min of recording was used? Was the minute chosen randomly selected?

Authors: Yes, it was selected randomly; the information is added.

L187: there are multiple ways to measure movement (stop-start activity, acceleration etc.). Please specify the type of movement that was measured, e.g., distance over time (cm/min).

Authors: Yes, it was the distance over time in cm/min, and the information is now provided here as well.

L190: How long was the UV exposure period? How many is ‘a group’?

Authors: The experimental details are now described in detail in section 2.3, and the specific sample sizes are added to the figure captions as requested below.

L208-212: some explanation of the unit (tail moment) is warranted given that this is not assumed knowledge for the generalist readership of biology journal.

Authors: Sure, it’s now added.

Section 3.1 (L257-284): were experimental UV treatments designed after considering these environmental measurements? If this is the case, then the authors have done a fine job in attempting to maximise the ecological relevance of their experiment. However, it seems odd to describe these environmental measurements after the experimental levels and not before. In my opinion (this is optional), this section would make more sense at L149, before the experimental treatments are described. That way the reader is more aware of how the experimental UV conditions were chosen.

Authors: It’s of course tempting to say that such experimental conditions were chosen in a strict way from the beginning, but in reality initially we made only some preliminary comparisons between the lamp UV levels and the solar UV in our region in order to make sure that experimental exposures are somewhat environmentally realistic. More quantitative measurements were made already when the experiments started as it took time to figure out how we can make underwater UV estimations. However, since the experimental UV levels were known to be lower than natural solar ones from the beginning, it was just a matter of what specific depths we modeled with the chosen experimental levels. Regarding the section 3.1, we consider this (maybe small but) an important result of the study and potentially interesting for other researchers since we could not find any openly available data of UV monitorings specifically in this format for our region. Thus, since we treat these data as a result and not just supplementary tests, we should place it in the Results and not Methods.

L260, L263: when referring to UVA and UVB levels (W/m2), the correct term is irradiance, not illuminance.

Authors: Thank you for this remark! We indeed previously mixed the terms up, but now it’s corrected along the whole text. In particular, UV is always presented as irradiance, solar visible light is now as well converted to irradiance, while visible light from the lamps is still presented as illuminance since the process of conversion in their case is not so straightforward and illuminance makes more sense here.

L267-L268: I am unsure of how UVC was detected in the field given that it is fully absorbed by the Earth’s atmosphere. Furthermore, I am unsure of the ecological relevance of the statement and I would consider removing it.

Authors: After the criticism from Reviewer 1, as well as your question about the specific spectrum ranges for channels of the UV-radiometer, we figured out that the device is not designed for UV-C measurements from the sun and sun-mimicking sources since the UV-C channel is also slightly sensitive to UV-B (in range 280-290 nm). Since all the obtained "UV-C" readouts are clearly caused by UV-B, we removed those measurements from the text and Table S1.

L278-279: Was this percentage including or excluding the effect of the glass on the sensor?  Please clarify in text.

Authors: The value 36.5±2.3 % is the decrease in total UV-A+UV-B irradiance as measured by the probe (this radiometer doesn’t differentiate UV-A and UV-B) on the shoreline and under 1 m of water. We are unsure how the raw measurements can be corrected for (1) the effect of the glass and (2) the angle between the axis of the probe and the sun. Thus, we directly write earlier in this paragraph that the estimate is only relative between 0 and 1 m depths. Table S1 contains all raw data and calculations for these measurements.

L279: Table S1 should be clearly labelled as such within the supplementary file.

Authors: Pardon us; we indeed forgot to include the description of Table S1 into the main text. It’s now added (if the file is not available during review please check the comment to L515 below).

L286-289: methods should not be repeated in the results section, I suggest deleting. Same here: L327-328.

Authors: We shortened both text fragments in order to provide only the essential context and removed excessive technical details.

L292: this is the first mention of the duration of the experiment, which should be stated in the experimental design section of the methods.

Authors: It’s now corrected as suggested.

L300: Have the authors considered a statistical comparison between UV-A and UV-B treatments for each species? Or statistical comparison between deepwater species? This could provide statistical evidence to the claims that O. albinus is more sensitive than O. flavus.

Authors: The UV-A and UV-B treatments included different levels of visible light, so we cannot differentiate specifically the factor of UV in such comparison. However, the comparisons between deep-water species are indeed interesting and we’ve added them to the text as suggested: «Moreover, O. flavus showed longer survival than O. albinus in both experiments with estimated LT50 of 8.5 versus 5 days under UV-A lamps and 8 versus 4 days under UV-B lamps. The greater mortality of O. albinus under both UV types is also supported by the statistically significant differences in direct comparisons between the survival curves of two deep-water species (both p < 0.001)». Thank you for this suggestion!

After this comment we’ve also had some hesitation about applying the correction for multiple comparisons to the survival data. I.e., the correction is usually applied within the same axes (facet of a figure panel in our case), but in this study we, in fact, now report more than one comparison per facet. Within one figure panel (i.e. within one type of UV treatment) we have: two comparisons of standard control groups with dark control groups for the deep-water species (not presented on the figures but we mention them in the Methods and now added the p-values there), four main comparisons of experimental with the main control groups and now also one comparison of the experimental groups of the deep-water species. This is seven comparisons per four facets. We’ve tried to apply the Holm’s correction to these seven multiple comparisons (which may be considered as an overkill by some researchers, yet we were interested in the result) but all the uncorrected significant p-values are initially small enough, so the correction doesn’t change anything even under these conditions in both UV types. However, the corrected p-values for the littoral species might be considered overcorrected by some readers with such treatment, which can be misleading in interpreting the overall results.

Thus, we decided to keep the mentioned p-values exactly as they were presented originally without the correction.

L305-307: This information needs to be provided in the methods instead. Same for other parts of the results section: L328-331, L342-344, L352-355, L372-377.

Authors: We ensured that all the information given in these text fragments in the previous manuscript version is now clearly described in the Methods and shortened most of them in the Results.

L310: please clarify in text what comparisons the two p values are referring to.

Authors: Added.

L382-385: Were control treatments conducted (each species living separately under the same conditions)? Without this control, it is not possible to determine whether co-inhabitation of O. flavus and E. cyaneus affected their survival, or if it was just background mortality.

Authors: Unfortunately, we didn’t include such control groups. However, we consider co-inhabitation of O. flavus with E. cyaneus (showing good survival of O. flavus for a three-weeks experiment) as kind of "control" treatment of O. flavus with the smaller animal, which is unable to prey on the larger O. flavus, to the "main" exposure of O. flavus with E. verrucosus of comparable size. In fact, seeing how quickly Ommatogammarus scavengers are able to devour other amphipods in deep-water traps, we expected O. flavus to prey on Eulimnogammarus, not vice versa. Thus, we consider even such a small experiment to clearly show that it is not the case and we make any conclusions only about the role of E. verrucosus at the very end of Discussion. Our motivation should probably be clearer with adding the following sentences to Introduction after earlier suggestions from you and Reviewer 2: «Besides evolutionary significance, UV intolerance of deep-water animals might have ecological implications for predicting their migration patterns if the lake water becomes less transparent for solar UV»… «Additionally, we tested the ability of the deep-water species to prey on the littoral amphipods without the pressure of solar UV radiation to evaluate their potential to invade more shallow-water communities if the lake water becomes less transparent».

L449-451: To confirm the theory that the treatments did not induce significant ROS production would require the direct measurement of ROS or the damage it causes (e.g., protein carbonyl and lipid peroxidation). These future directions should be recognised in text.

Authors: We now softened the conclusion in this paragraph and added the following comment as suggested: «Importantly, more direct markers of elevated ROS production such as lipid peroxidation products or protein carbonylation were not estimated in this study and are necessary to draw the final conclusions». In fact, we tried to estimate the lipid peroxidation products in part of the frozen samples but could not use the measurements due to methodological problems. This is why we have small sample sizes for some groups in the enzymatic data, in case you might we wondering.

L454-L465: These are all plausible explanations, but it is important to acknowledge in the discussion that this assay did not measure the primary type of DNA damage caused by UV radiation.

Authors: It is now added as requested.

L498-500: this last sentence is confusing. Maybe change “would not probably” to “would be unlikely to”.

Authors: Corrected; thank you.

L515: I am not sure what Table S1 is referring to.

Authors: We’ll again upload updated Table S1 containing all raw data to this submission as supplementary material, but we are not sure that the MDPI system will show it to you. If this is the case, please check the Supplementary Material section of the first manuscript version presented as preprint at https://www.preprints.org/manuscript/202410.0306. The file preprintsSupplementary202410.0306.v1 is downloaded for us in BIN format for some reason, but MS Excel and other spreadsheet editors open it with no problem. The updated Table S1 is almost identical to this version except for no UV-C information and added solar irradiance converted from illuminance in the UV monitoring sheet, as well as slightly corrected day labeling for the locomotor activity and added raw data to Fig 8 since the numbers were removed during re-building the figure.

Figures

Figure 2 could probably go into supplementary material.

Authors: To our opinion, Figure 2 is important in the main text for two reasons: (1) to clearly show the natural levels of UV and readily emphasize ecological relevance of the used exposure levels and (2) to demonstrate good correlation between UV-A and UV-B despite more poor relation to visible light. The latter additionally demonstrates that complete filtering of UV-B from UV-A exposures and vice versa is not critical for experimental exposures. Furthermore, all raw data for the UV monitoring are already presented in Table S1.

Some general tips for tidying all the figures include (1) placing units in brackets instead of separating with a comma and (2) removing gridlines. Figure panels should be labelled (a), (b) and so on to make it easier to refer the reader to specific graphs within the figures. Figure legends also need to define whether horizontal lines in some figures represent mean or median.

Authors: Corrected as suggested: (1) units are now in brackets; (3) figure panels are labeled with letters; (4) figure captions include description of the horizontal lines (they are all medians with no exceptions). (2) We’ve also tried to remove the grid lines as suggested but in this case the multi-facet panels (in Figs 3, 4, 5, 6, 8) visually lose the borders between the facets; lines and boxplots just "hang in the air" and the data are difficult to perceive, especially for the mortality lines. Thus, we would like to keep this visual style unified over all plots.

Figure 2: it may be worthwhile including a line of best fit on panels C and D.

Authors: Added. Additionally, we replaced the correlation coefficients with coefficients of determination of the linear models since they seem to make more sense in this context.

There is an unusual ‘clear sky’ point with very low irradiance at 15:00. Consider whether there is a reasonable explanation for this value or whether it should be considered a potential error/outlier for removal?

Authors: We’ve checked the raw data, and the point indeed was mislabeled. Now it’s corrected; thank for noticing.

Figure 2 caption (L270-273): This figure caption is repetitive and can be simplified, e.g., UV-A (a) and UV-B (b) intensities corresponding to the 24-hour time format.

Authors: Corrected as suggested.

Figure 3: Please specify in the figure caption what the coloured shading around the solid survival trend lines indicate.

Authors: The shadings are the 95 % confidence intervals for the Kaplan-Meier survival curves; the information is now added.

Figure 4: There are some major points of unclarity with this figure and the locomotor performance data. Firstly, it is not clear what samples sizes are being worked with, and whether mortality is affecting this. Presumably, the lack of data for O. albinus after day 6 is due to mortality. Sample size ranges should be provided in figure captions or described in the methods.

Authors: The sample sizes are now indicated throughout the figure captions. Additionally, the fact of mortality affecting the sample sizes is now clearly acknowledged in the Methods.

Secondly, the x-axis is very random because it misses day 4 and 5, and then jumps to a 2-day interval. Why were some days excluded from the figure?

Authors: We did not exclude any obtained data from the figure. The presented time points were indeed chosen initially to present the beginning of the experiment every day and then switch to the 2-day interval since high frequency is not required on the longer term. We’ve also just found a mismatch in day labeling on the x-axis after your comment, so please don’t be confused by slightly shifted day numbers on the updated figure, and thank you for the comment!

Figure 5, 6 and 7: Some form of error also needs to be shown (e.g., standard error, standard deviation, confidence interval etc.).

Authors: The standard way of presenting the distribution of the data that were analyzed with non-parametrical tests is boxplots showing such statistics as minimum, median, maximum and quartiles between them. However, when the number of replicas is relatively low, it’s also very common to just draw the median and all raw data points, which basically show the same statistics but in more fair way for such a case. Here we follow the recommendations from, for example, the review by Weissgerber et al. 2015 in PLOS Biology (doi: 10.1371/journal.pbio.1002128). Thus, we believe it’s more relevant to draw the actual data distribution than boxplots on the mentioned figures.

Figure 8: This time-series data is better presented as a line graph

Authors: This is fair criticism, thank you. The figure is now re-built, and the raw data are moved to Table S1.

Authors: Thank you very much again for your work and the provided suggestions!
